# The genetic architecture of appendicular lean mass characterized by association analysis in the UK Biobank study

Yu-Fang Pei [1,2,6,7✉], Yao-Zhong Liu[3,6], Xiao-Lin Yang [4,6], Hong Zhang[2,5], Gui-Juan Feng[1,2], Xin-Tong Wei[1,2] & Lei Zhang [2,5,7✉]

Appendicular lean mass (ALM) is a heritable trait associated with loss of lean muscle mass and strength, or sarcopenia, but its genetic determinants are largely unknown. Here we conducted a genome-wide association study (GWAS) with 450,243 UK Biobank participants to uncover its genetic architecture. A total of 1059 conditionally independent variants from 799 loci were identified at the genome-wide significance level ($p < 5 \times 10^{-9}$), all of which were also significant at $p < 5 \times 10^{-5}$ in both sexes. These variants explained ~15.5% of the phenotypic variance, accounting for more than one quarter of the total ~50% GWAS-attributable heritability. There was no difference in genetic effect between sexes or among different age strata. Heritability was enriched in certain functional categories, such as conserved and coding regions, and in tissues related to the musculoskeletal system. Polygenic risk score prediction well distinguished participants with high and low ALM. The findings are important not only for lean mass but also for other complex diseases, such as type 2 diabetes, as ALM is shown to be a protective factor for type 2 diabetes.

[1] Department of Epidemiology and Health Statistics, School of Public Health, Medical College of Soochow University, Soochow, Jiangsu, PR China. [2] Jiangsu Key Laboratory of Preventive and Translational Medicine for Geriatric Diseases, School of Public Health, Medical College of Soochow University, Soochow, Jiangsu, PR China. [3] Department of Biostatistics and Data Science, Tulane University School of Public Health and Tropical Medicine, New Orleans, LA, USA. [4] Department of Research, The Affiliated Hospital of Yangzhou University, Yangzhou University, Yangzhou, Jiangsu, PR China. [5] Center for Genetic Epidemiology and Genomics, School of Public Health, Medical College of Soochow University, Soochow, Jiangsu, PR China. [6]These authors contributed equally: Yu-Fang Pei, Yao-Zhong Liu, Xiao-Lin Yang. [7]These authors jointly supervised this work: Yu-Fang Pei, Lei Zhang. ✉email: ypei@suda.edu.cn; lzhang6@suda.edu.cn

Lean body mass is an important physiological index. Low lean body mass, together with low muscle strength and low physical performance, represents a key component for the definition of sarcopenia, which is a critical condition with functional impairment and physical disability and a major modifiable cause of frailty in the elderly[1,2]. Lean body mass is associated with bone mineral density and hence may also influence the risk for osteoporosis[3]. Other lean body mass-related conditions include dysmotility syndrome[4], sarcopenic obesity[5] and cachexia[6].

Lean body mass has a significant genetic component, as evidenced by a high heritability of 50–80% observed in twin studies[7,8]. However, findings of specific genes for human lean mass variation remain limited, even with the powerful genome-wide association study (GWAS) approach. A key reason for the limited findings, as in other human complex traits, is the modest sample size used in most GWASs performed for lean body mass[9–13], resulting in few single nucleotide polymorphisms (SNPs) identified with genome-wide significance.

As a notable example, one previous large meta-analysis of GWASs amassed 20 cohorts of European ancestry with a total sample size of >38,000 for whole body lean mass and of >28,000 for appendicular lean mass (ALM)[14]. Of the 20 studied cohorts, 10 ($n = 21,074$, 55%) were characterized with the dual-energy X-ray absorptiometry (DEXA)-derived measure of lean mass, while the other 10 ($n = 17,218$, 45%) were characterized with the bioelectrical impedance analysis (BIA)-derived measure. Despite the large sample used, the percentage of phenotypic variance explained by the identified SNPs was still only 0.23% and 0.16% for whole body lean mass and ALM, respectively, suggesting that most of the heritability of lean body mass was still undetected. Therefore, even with such a large GWAS meta-analysis, it is still necessary to boost the sample size further to enhance the statistical power for detecting more causal SNPs underlying lean body mass.

In a more recent study, a GWAS of ALM was conducted in a population of 85,750 middle-aged (aged 38–49 years) individuals from the UK Biobank (UKB)[15]. A total of 182 loci were identified, 78% of which were replicated in a population of 181,862 elderly (aged 60–74 years) individuals from the same UKB cohort.

Unlike ALM, which is mainly affected by skeletal muscle, whole body lean mass is determined by skeletal muscle, smooth muscle and cardiac muscle. Therefore, ALM has a higher predictive power for sarcopenia-related health outcomes because sarcopenia is mainly due to a low skeletal muscle amount. ALM is also slightly more heritable than whole body lean mass[16] and as such a more suitable trait for sarcopenia-related genetic analyses.

Here, in this study, with a sample containing approximately half-million participants of European origin, we performed a GWAS of ALM in the full UKB cohort. At a stringent genome-wide significance level ($p < 5 \times 10^{-9}$), we identified >1000 independent variants that were significant, all of which were significant in both sexes at $p < 5 \times 10^{-5}$. Our findings revealed a large number of genetic variants for lean body mass and contributed to the characterization of the genetic architecture of this complex trait. Through this GWAS, we demonstrated the power for mapping the genetic landscape of common human complex traits/diseases using extraordinarily large samples.

## Results

A flow chart of this study is displayed in Fig. 1. The study sample came from the UKB cohort, which is a large prospective cohort of ~500,000 participants from across the United Kingdom, aged between 48 and 73 at recruitment. The basic characteristics of the sample are listed in Supplementary Data 1. In this study, we quantified ALM by appendicular fat-free mass measured by BIA. This measurement of lean mass is reliable based on its strong correlation with ALM measured by DEXA in 4294 UKB participants (Pearson's correlation coefficient 0.96, $p < 2.2 \times 10^{-16}$). The Pearson correlation coefficient between ALM and AFM was 0.68 in males and 0.79 in females. Neither ALM nor AFM followed a normal distribution. Instead, both distributions were right skewed (Supplemental Fig. 1).

ALM for all eligible participants was adjusted by AFM and other covariates, and the residuals were transformed into a standard normal distribution so that no outliers were observed in the transformed phenotype $ALM_{adj}$.

**Main association results**. Following QC of both $ALM_{adj}$ and genome-wide genotypes, data from 19.4 million variants with a minor allele frequency (MAF) > 0.1% and an imputation quality score >0.3 were available in 244,730 female and 205,513 male participants.

In each sex group, the additive effect of each variant was tested on $ALM_{adj}$ with BOLT-LMM[17] (Fig. 2). The genomic inflation factor showed notable inflation in both sex groups ($\lambda_{female} = 2.09$, $\lambda_{male} = 1.84$). The LDSC mean Chi-square and intercept were 3.05 and 1.17 for females and 2.63 and 1.13 for males, corresponding to attenuation ratios of 0.083 and 0.077, respectively.

LDSC estimated a genetic correlation coefficient as high as 0.93 (s.e. 0.01) between the two sexes, implying that most GWAS-attributable heritability was shared between sexes.

Given the shared heritability between sexes, between-sex meta-analysis was performed with an inverse-variance-weighted fixed-effects model to combine the sex-specific GWAS results. The meta-analysis signals had an attenuation ratio of 0.078 (mean Chi-square = 4.50, intercept = 1.27), which was equal to that estimated by analysing the subset of genetically determined "Caucasian" participants ($N = 400,879$, attenuation ratio = 0.078), suggesting a limited effect of population structure. However, some residual stratification may remain uncorrected because ancestry was based upon self-reporting.

A total of 121,109 variants were significant at the genome-wide significance (GWS, $\alpha = 5.0 \times 10^{-9}$) level in the combined meta-analysis and were significant at $p < 5 \times 10^{-5}$ in both sexes. Based on their physical coordinates, these variants were divided into 828 loci that were at least 500 kb apart. It was found that 57 lead variants were not in complete linkage equilibrium (LD $r^2 < 0.1$) with each other due to the long-range LD pattern. After removing 33 lead variants, the remaining 795 lead variants were all in linkage equilibrium. Therefore, they were treated as independent loci.

Approximate conditional association analysis followed by between-sex meta-analysis was recursively performed, which further identified 4 additional loci and 264 conditionally significant variants at the GWS level that were also significant in both sexes at $p < 5 \times 10^{-5}$. These additional variants were also in linkage equilibrium (LD $r^2 < 0.1$) with the 795 primary lead variants.

In total, 1059 (i.e. $795 + 264$) independent variants from 799 distinct loci were associated with $ALM_{adj}$ (Supplementary Data 2). Among them, 353 achieved the strongest significance level ($p < 5 \times 10^{-9}$) in both sexes (categorized here as Tier 1 variants). Additionally, 208 variants achieved $p$ values $<5 \times 10^{-9}$ in females and $p$ values $<5 \times 10^{-5}$ in males; 94 variants achieved $p$ values $<5 \times 10^{-9}$ in males and $p$ values $<5 \times 10^{-5}$ in females (categorized here as Tier 2 variants). Finally, 404 variants achieved $p$ values $<5 \times 10^{-5}$ in both sexes and $p$ values $<5 \times 10^{-9}$ in the between-sex meta-analysis (categorized here as Tier 3 variants).

**Replication in UKB South Asian participants**. The associations of the 1059 lead SNPs in the 7,452 UKB South Asian (Indian,

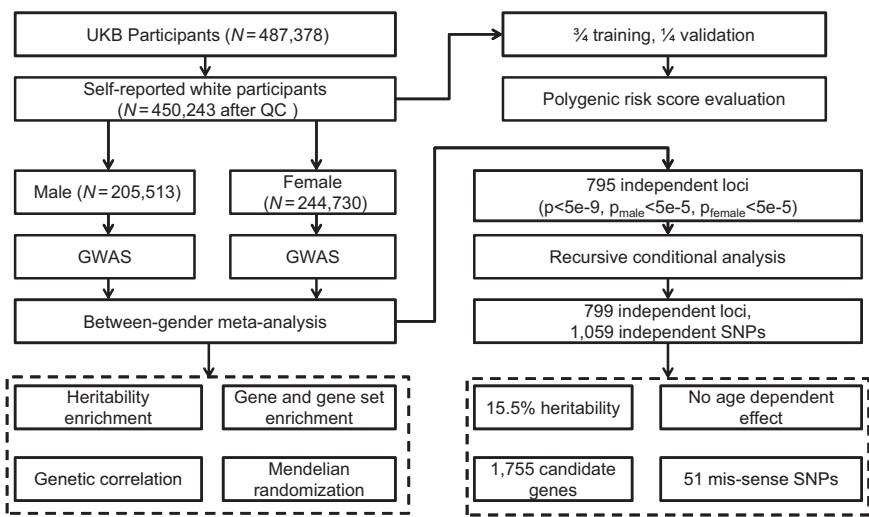

**Fig. 1 Flow chart of the present study.** The study sample came from the UKB cohort. After QC, 450,243 self-reported white participants were eligible for analysis. The two sex groups were analyzed separately and then were combined for a meta-analysis. A series of in-depth analyses, including heritability enrichment, gene and gene sets enrichment, genetic correlation, Mendelian randomization, and polygenic risk score profiling were performed to uncover the genetic architecture of the trait.

Pakistani and Bangladeshi) participants are listed in Supplementary Data 3. At the stringent significance level of $4.72 \times 10^{-5}$ (0.05/1,059), only one SNP, rs4338565, was significant ($p = 2.30 \times 10^{-5}$), which is not unexpected given the limited sample size. At the nominal level of 0.05, 124 SNPs were significant. Overall, the effect direction at 767 SNPs was consistent between the two analyses (binomial $p < 2.2 \times 10^{-16}$), and the correlation coefficient of the regression between them was 0.27 (95% CI [0.22, 0.33], $p < 2.2 \times 10^{-16}$), partially demonstrating the replicability in the smaller South Asian sample.

**Replication of previously reported loci.** Associations of previously reported SNPs in the present study are listed in Supplementary Data 4. Among them, all five lead SNPs reported by Zillikens et al.[14] were successfully replicated at the nominal level of $p < 0.05$, and up to four of them were replicated at the $5.0 \times 10^{-8}$ level. For another bivariate study in which eight loci were identified[18], two were significant at the $5.0 \times 10^{-8}$ level, five additional were significant at the nominal level, but the last SNP rs7672749 was not significant ($p = 0.26$). For the recent study of ALM in middle-aged and older UKB participants, all 182 lead SNPs were nominally significant in the present study, and up to 168 (92.3%) were significant at the GWS level. In sum, the present study well replicates the findings from previous smaller studies.

Of the 799 identified loci, 102 were reported in the above-mentioned previous studies, while the remaining 697 were not.

**Overlap with loci for obesity-related traits.** We also evaluated the overlap of the identified loci with those identified for three obesity traits obtained through the GIANT consortium. Flanking variants surrounding 478 lead SNPs (defined as the lead SNP + 500 kb flanking region to either side) were associated with one or more obesity traits at the $5.0 \times 10^{-8}$ level, while no flanking variants surrounding the remaining 581 lead SNPs showed associations at the $5.0 \times 10^{-8}$ level, demonstrating their novelty and possible specificity to lean but not fat mass.

**Sex specificity.** Our analysis identified an additional 107 loci that were significant at the GWS level in the between-sex meta-analysis but not significant at the suggestive level ($p < 5 \times 10^{-5}$) in

each sex group (Supplementary Data 5). These loci may represent sex-specific signals pending further replication.

**Sex heterogeneity.** Of the 1059 identified variants, 181 (17.1%) exhibited high between-sex meta-analysis heterogeneity ($I^2 > 50\%$), most (175) of which were Tier 1 or 2 variants. A statistical test of sex differences in genetic effects revealed no difference in any SNP after accounting for multiple testing ($\alpha = 0.05/1,059 = 4.72 \times 10^{-5}$), suggesting that all of the identified variants had similar effect sizes between sexes.

**Age-dependent effect.** The 1059 lead variants were evaluated for their age-dependent effects. At the Bonferroni-corrected significance level ($p < 0.05/1,059 = 4.72 \times 10^{-5}$), no age-dependent effects were identified at the 1059 lead SNPs (Supplementary Data 6). The most significant hit was rs2310876 ($p = 2.01 \times 10^{-3}$), where the per allele effect became larger as age increased. The second strongest hit was rs550793660 ($p = 3.48 \times 10^{-3}$). In contrast, the allele effect at this SNP tended to become zero as age increased.

**Heritability distribution.** The 1059 identified variants included 938 common (MAF > 5%), 97 less common (5%≥MAF > 1%) and 24 rare (MAF ≤ 1%) variants. Variants with a smaller MAF generally had a larger per allele effect size (Fig. 2). For example, the average per allele effect size of rare variants (mean 0.14, s.d 0.08) was 7-fold larger than that of common variants (mean 0.02, s.d 0.009).

The total phenotypic variance explained by the 1059 lead variants was 17.8% before correction for the winner's curse effect and 15.5% after the correction. Using BOLT-REML[19], GWAS-attributable total heritability was estimated to be 0.486 (s.e $3.14 \times 10^{-3}$) and 0.476 (s.e $3.63 \times 10^{-3}$) in females and males, respectively. After removing all variants in the flanking 500 kb regions surrounding each of the 1059 lead variants from raw genotypes, the remaining heritability was estimated to be 0.207 (s.e $2.94 \times 10^{-3}$) and 0.188 (s.e $3.34 \times 10^{-3}$) in females and males, respectively, implying that the identified loci collectively explained heritabilities of 0.279 and 0.288 in females and males, respectively. Therefore, ~10% of the heritability at the identified loci remains undefined.

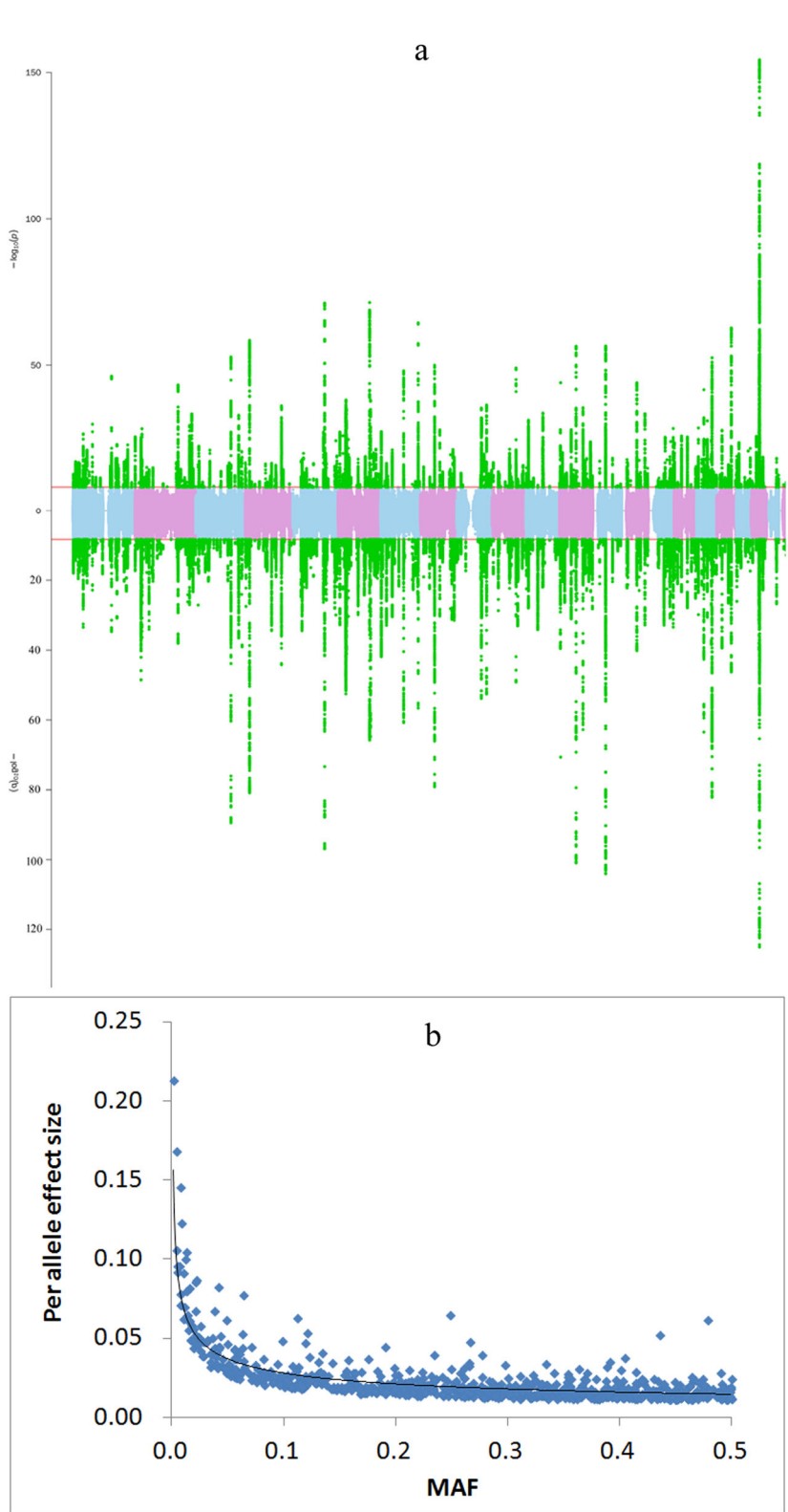

**Fig. 2 Main association results. a** Manhattan plot of the meta-analyses of both sexes (males: top, females: bottom). The horizontal red line indicates the genome-wide significance level ($\alpha = 5 \times 10^{-9}$) on the $-\log_{10}$ scale. All significant SNPs are marked in green. **b** Per allele effect size versus minor allele frequency (MAF). The X-axis is the MAF of the 1059 identified variants, and the Y-axis is the per allele effect size (regression coefficient). The black solid line is the power function fitted to the data.

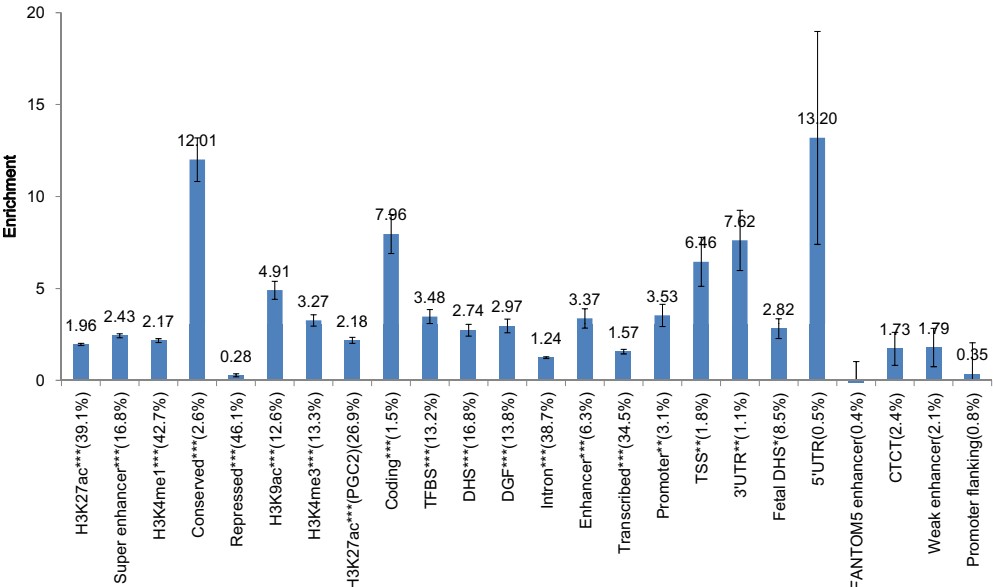

**Fig. 3 Heritability enrichment in different functional annotations.** Enrichment of genome-wide association signals in 24 main annotations using LDSC regression. The *Y*-axis represents the ratio of phenotypic variance explained by variants in a particular annotation category against that explained in the remaining regions. Error bars represent jackknife standard errors around the estimates of enrichment. A single asterisk indicates significance at *p* < 0.05 after Bonferroni correction for the 24 hypotheses tested, and two asterisks indicate significance at *p* < 0.01.

Applying the stratified LDSC analysis, the GWAS-attributable heritability was partitioned into 24 functional categories[20]. Statistically significant enrichment was observed for 19 functional categories ($p < 0.05/24 = 2.08 \times 10^{-3}$, Fig. 3). In line with the observations by Finucane et al.[20], regions conserved in mammals showed the strongest enrichment of any category, with 2.6% of SNPs explaining an estimated 31.3% of SNP heritability (enrichment ratio = 12.0, $P = 1.48 \times 10^{-18}$). Other categories with significant enrichment included coding regions (enrichment ratio = 8.0, $P = 4.10 \times 10^{-10}$), 3′ UTRs (enrichment ratio = 7.6, $P = 9.29 \times 10^{-5}$), transcription start sites (enrichment ratio = 6.5, $P = 6.07 \times 10^{-5}$), and H3K9ac histone marks (enrichment ratio = 4.9, $P = 1.02 \times 10^{-13}$). Neither the promoter nor the 5′-UTR region showed significant enrichment, although the 5′-UTR region had a high estimate of the enrichment ratio (13.2, $p = 0.04$).

Using a new function of the stratified LDSC method[21], three tissues were enriched at a Bonferroni-corrected significance level ($P < 0.05/205 = 2.44 \times 10^{-4}$), including cartilage ($p = 1.01 \times 10^{-7}$), chondrocytes ($p = 9.48 \times 10^{-5}$) and uterus ($2.02 \times 10^{-4}$).

**Candidate gene prioritization**. To prioritize candidate genes at the associated loci, we used multiple analytical strategies. A set of credible risk variants (CRVs) at each locus were defined as variants with high LD with the lead variant ($r^2 > 0.8$). A total of 27,988 CRVs were identified (Supplementary Data 7). Based on these CRVs, six sources of supporting evidence were used to prioritize 1755 candidate genes (Supplementary Data 8–12).

A number of genes had multiple lines of supporting evidence (Supplemental Fig. 2). The *IGFBP3* (IGF-Binding Protein 3) gene at 7p12.3, for example, had four lines of supporting evidence. This gene was closest to the lead SNP chr7:46262082 at the locus. One CRV, rs11977526, was associated with its mRNA expression level in skeletal muscle ($p = 3.21 \times 10^{-5}$) and associated with its protein level ($p = 3.80 \times 10^{-5}$) in whole blood. Finally, it was prioritized by DEPICT ($p = 5.66 \times 10^{-4}$). Another example is *CXXC5* (CXXC Finger Protein 5) at 5q31.2. The lead SNP rs3822742 is an intron variant in *CXXC5*. Multiple CRVs around rs3822742, such as rs356452, were associated with *CXXC5* mRNA

expression in whole blood samples. This gene was also prioritized by both DEPICT and SMR.

**Comparison between imputation- and sequencing-based association signals**. Of the 1059 identified variants, 51 were missense coding variants. They are also available in the recently released UKB exome-sequencing data that contain a subset of ~50,000 participants from the whole UKB cohort. Using a set of 45,554 unrelated European participants who were both genotyped/imputed and sequenced, we compared the imputation-based association results with exome-sequencing-based results. Raw genotypes were highly concordant between the imputed and sequenced data ($r^2 = 0.84–1.00$). The imputation-based and sequencing-based *p* values were highly concordant (Supplementary Data 13). For example, the imputation-based *p* values were within a 2-fold difference of the sequencing-based *p* values for up to 48 variants. Overall, these observations support that imputation-based association signals are close to the real sequencing-based association signals in a large sample. Therefore, imputation-based GWAS may be able to identify true associations, even those of rare variants.

**Missense variants and the associated genes**. As mentioned above, of the 1059 identified variants, 51 were missense coding variants. The majority (28) of these 51 missense mutations were predicted to be deleterious by one or more bioinformatics tools, including PolyPhen-2[22], SIFT[23], PROVEAN[24] and FATHMM[25] (Supplementary Data 14), supporting their potential functional relevance.

Missense mutations were enriched among rare variants. Eight of the 24 rare variants were missense mutations, in clear contrast to the 43 missense mutations among the remaining 1035 variants (odds ratio = 11.53, Fisher's exact test $p = 6.77 \times 10^{-6}$). Evidence of the enrichment was strengthened by comparing 24 missense mutations from 121 rare or less common variants with 27 missense mutations from 938 common variants (odds ratio = 6.89, Fisher's exact test $p = 1.10 \times 10^{-9}$), suggesting that low-

**Table 1 Association results of 51 missense variants.**

| RSID | CHR | POS | EA/OA | FRQ | Gene | AA | Condition | Male (N = 205,513) | | | Female (N = 244,730) | | | Meta-analysis (N = 450,243) | | | | |
|---|---|---|---|---|---|---|---|---|---|---|---|---|---|---|---|---|---|---|
| | | | | | | | | BETA | SE | P | BETA | SE | P | BETA | SE | P | I²(%) | P_het |
| *Rare* | | | | | | | | | | | | | | | | | | |
| rs143365597 | 1 | 41540902 | A/G | 0.004 | SCMH1 | Ser313Pro | S | 0.12 | 0.025 | 3.74E-06 | 0.16 | 0.021 | 1.22E-13 | 0.14 | 0.016 | 5.62E-18 | 42.2 | 0.19 |
| rs148330006 | 1 | 86048526 | C/G | 0.992 | CCN1 | Ser316Cys | S | 0.07 | 0.015 | 5.28E-06 | 0.07 | 0.013 | 2.00E-07 | 0.07 | 0.010 | 4.83E-12 | 0.0 | 0.94 |
| rs7318210 | 3 | 169831268 | T/C | 0.993 | PHC3 | Lys745Glu | P | -0.10 | 0.017 | 5.20E-09 | -0.08 | 0.015 | 2.50E-07 | -0.09 | 0.011 | 5.63E-15 | 0.0 | 0.38 |
| rs78408340 | 5 | 102338739 | C/G | 0.991 | PAM | Ser539Trp | P | 0.09 | 0.015 | 7.30E-09 | 0.07 | 0.013 | 4.50E-09 | 0.08 | 0.010 | 5.98E-16 | 0.0 | 0.55 |
| rs148833559 | 5 | 172755066 | A/C | 0.001 | STC2 | Leu44Arg | P | 0.38 | 0.038 | 6.70E-25 | 0.38 | 0.034 | 1.10E-30 | 0.38 | 0.025 | 9.21E-52 | 0.0 | 0.98 |
| rs112892337 | 8 | 135614553 | C/G | 0.004 | ZFAT | Ser470Cys | S | 0.16 | 0.021 | 4.53E-14 | 0.13 | 0.019 | 2.32E-11 | 0.14 | 0.014 | 1.23E-23 | 15.7 | 0.28 |
| rs141308595 | 15 | 89424870 | T/G | 0.002 | HAPLN3 | Ser133Arg | S | -0.18 | 0.036 | 9.33E-07 | -0.17 | 0.034 | 4.19E-07 | -0.17 | 0.025 | 1.84E-12 | 0.0 | 0.88 |
| rs35816944 | 16 | 1828030 | A/G | 0.007 | SPSB3 | Leu171Ser | S | -0.12 | 0.016 | 4.94E-14 | -0.10 | 0.015 | 4.28E-11 | -0.11 | 0.011 | 2.49E-23 | 17.5 | 0.27 |
| *Less common* | | | | | | | | | | | | | | | | | | |
| rs6540718 | 1 | 212274413 | A/C | 0.014 | DTL | Lys694Thr | S | 0.07 | 0.012 | 9.86E-10 | 0.04 | 0.010 | 3.32E-05 | 0.05 | 0.008 | 1.16E-12 | 75.2 | 0.04 |
| rs41265094 | 2 | 227661003 | C/G | 0.985 | IRS1 | Arg818Gly | S | -0.05 | 0.011 | 1.36E-05 | -0.08 | 0.010 | 1.53E-14 | -0.06 | 0.007 | 1.06E-17 | 78.2 | 0.03 |
| rs139921635 | 4 | 73181637 | T/G | 0.024 | ADAMTS3 | Thr513Pro | S | 0.04 | 0.010 | 1.93E-05 | 0.04 | 0.008 | 3.21E-06 | 0.04 | 0.006 | 2.66E-10 | 0.0 | 0.84 |
| rs33966734 | 6 | 41903798 | A/C | 0.014 | CCND3 | Asp253Glu | S | -0.11 | 0.012 | 5.85E-21 | -0.11 | 0.011 | 3.23E-22 | -0.11 | 0.008 | 1.89E-41 | 0.0 | 0.58 |
| rs62621812 | 7 | 127015083 | A/G | 0.020 | ZNF800 | Ser103Pro | P | 0.06 | 0.010 | 5.20E-10 | 0.08 | 0.009 | 1.50E-20 | 0.07 | 0.007 | 3.16E-27 | 63.6 | 0.10 |
| rs17480616 | 7 | 135123060 | C/G | 0.031 | CNOT4 | Ala7Gly | S | 0.04 | 0.008 | 4.28E-08 | 0.04 | 0.007 | 1.87E-06 | 0.04 | 0.005 | 4.50E-13 | 0.0 | 0.56 |
| rs61688134 | 12 | 22017410 | T/C | 0.014 | ABCC9 | Ile734 Val | P | -0.06 | 0.012 | 3.80E-06 | -0.07 | 0.011 | 2.30E-10 | -0.06 | 0.008 | 7.88E-15 | 0.0 | 0.43 |
| rs61754233 | 12 | 77439823 | C/G | 0.020 | E2F7 | Pro275Arg | P | 0.04 | 0.010 | 2.26E-05 | 0.04 | 0.009 | 2.45E-06 | 0.04 | 0.007 | 2.36E-10 | 0.0 | 0.90 |
| rs28929474 | 14 | 94844947 | T/C | 0.020 | SERPINA1 | Lys366Glu | S | 0.05 | 0.010 | 2.60E-07 | 0.05 | 0.009 | 7.50E-06 | 0.05 | 0.007 | 1.05E-14 | 0.0 | 0.90 |
| rs78457529 | 16 | 24950880 | T/C | 0.012 | ARHGAP17 | Gln510Arg | P | -0.08 | 0.013 | 6.50E-10 | -0.10 | 0.012 | 5.00E-17 | -0.09 | 0.009 | 1.22E-24 | 0.0 | 0.39 |
| rs77542162 | 17 | 67081278 | A/G | 0.977 | ABCA6 | Cys1359Arg | P | -0.05 | 0.010 | 2.60E-06 | -0.07 | 0.009 | 1.70E-15 | -0.06 | 0.006 | 1.49E-19 | 66.0 | 0.09 |
| rs62621197 | 19 | 8670147 | T/C | 0.037 | ADAMTS10 | Gln62Arg | P | -0.04 | 0.008 | 1.50E-08 | -0.04 | 0.007 | 3.60E-11 | -0.04 | 0.005 | 1.11E-17 | 0.0 | 1.00 |
| rs34934920 | 19 | 38976655 | T/C | 0.025 | RYR1 | Leu1787Pro | P | -0.03 | 0.009 | 3.30E-05 | -0.04 | 0.008 | 5.70E-06 | -0.04 | 0.006 | 4.14E-09 | 0.0 | 0.85 |
| rs34093919 | 19 | 41117300 | A/G | 0.012 | LTBP4 | Asn752Asp | P | -0.06 | 0.013 | 8.20E-07 | -0.06 | 0.012 | 1.60E-07 | -0.06 | 0.009 | 2.64E-13 | 0.0 | 0.86 |
| rs4252548 | 19 | 55879672 | T/C | 0.022 | IL11 | His112Arg | P | -0.08 | 0.009 | 8.35E-18 | -0.08 | 0.008 | 6.79E-19 | -0.08 | 0.006 | 4.84E-35 | 0.0 | 0.69 |
| rs147110934 | 19 | 55993436 | T/G | 0.024 | ZNF628 | Asp292Glu | P | -0.09 | 0.009 | 2.80E-22 | -0.06 | 0.008 | 2.10E-13 | -0.07 | 0.006 | 9.39E-32 | 77.3 | 0.04 |
| *Common* | | | | | | | | | | | | | | | | | | |
| rs3850625 | 1 | 201016296 | A/G | 0.118 | CACNA1S | Cys1539Arg | P | -0.02 | 0.004 | 9.20E-06 | -0.03 | 0.004 | 9.00E-14 | -0.03 | 0.003 | 3.42E-18 | 33.9 | 0.22 |
| rs1260326 | 2 | 27730940 | T/C | 0.396 | GCKR | Leu446Pro | P | -0.03 | 0.003 | 5.30E-29 | -0.03 | 0.003 | 4.20E-37 | -0.03 | 0.002 | 6.16E-64 | 0.0 | 0.61 |
| rs867529 | 2 | 88913273 | C/G | 0.280 | EIF2AK3 | Ser136Cys | P | 0.01 | 0.003 | 1.60E-07 | 0.02 | 0.003 | 2.00E-14 | 0.02 | 0.002 | 1.00E-18 | 57.7 | 0.12 |
| rs1047891 | 2 | 211540507 | A/C | 0.316 | CPS1 | Asn1412Thr | P | 0.02 | 0.003 | 2.20E-09 | 0.03 | 0.003 | 6.30E-27 | 0.02 | 0.002 | 5.70E-31 | 85.5 | 0.01 |
| rs11545169 | 3 | 184020542 | T/G | 0.161 | PSMD2 | Asp313Glu | P | -0.04 | 0.004 | 3.30E-21 | -0.04 | 0.003 | 8.80E-25 | -0.04 | 0.003 | 8.19E-43 | 0.0 | 0.99 |
| rs1291602 | 5 | 130766662 | T/C | 0.159 | AC008695.1 | Gln1502Arg | P | -0.03 | 0.004 | 8.70E-16 | -0.03 | 0.003 | 3.80E-18 | -0.03 | 0.003 | 3.42E-32 | 0.0 | 0.82 |
| rs9379084 | 6 | 7231843 | A/G | 0.116 | RREB1 | Asn1171Asp | S | -0.03 | 0.004 | 2.76E-13 | -0.02 | 0.004 | 4.48E-09 | -0.03 | 0.003 | 1.84E-20 | 44.9 | 0.18 |
| rs1063582 | 8 | 23167353 | T/G | 0.235 | LOXL2 | Met570 Leu | P | 0.02 | 0.003 | 2.90E-05 | 0.02 | 0.003 | 1.40E-12 | 0.02 | 0.002 | 1.12E-16 | 46.0 | 0.17 |
| rs61729527 | 8 | 77761919 | T/C | 0.052 | ZFHX4 | Ser1273Pro | S | -0.04 | 0.006 | 7.73E-09 | -0.03 | 0.006 | 1.03E-08 | -0.03 | 0.004 | 4.53E-16 | 0.0 | 0.71 |
| rs10283100 | 8 | 120596023 | A/G | 0.056 | ENPP2 | Ser545Pro | P | -0.05 | 0.006 | 4.00E-20 | -0.06 | 0.006 | 7.40E-29 | -0.06 | 0.004 | 4.11E-44 | 0.0 | 0.56 |
| rs12541381 | 8 | 135649848 | A/G | 0.258 | ZFAT | Ser102Pro | P | -0.03 | 0.003 | 2.50E-19 | -0.03 | 0.003 | 3.30E-32 | -0.03 | 0.002 | 2.81E-49 | 48.6 | 0.16 |
| rs7916821 | 10 | 69933969 | A/G | 0.495 | MYPN | Asn707Ser | P | 0.02 | 0.003 | 3.68E-14 | 0.02 | 0.003 | 4.04E-09 | 0.02 | 0.002 | 3.79E-21 | 65.0 | 0.09 |
| rs2066827 | 12 | 12871099 | T/G | 0.768 | CDKN1B | Val109Gly | S | 0.02 | 0.003 | 2.58E-11 | 0.01 | 0.003 | 3.35E-07 | 0.02 | 0.002 | 1.76E-16 | 62.4 | 0.10 |
| rs2277339 | 12 | 57146069 | T/G | 0.896 | PRIM1 | Asp5Ala | P | 0.03 | 0.005 | 7.20E-15 | 0.04 | 0.004 | 5.00E-18 | 0.04 | 0.003 | 2.00E-30 | 0.0 | 0.80 |
| rs3764002 | 12 | 108618630 | T/C | 0.262 | WSCD2 | Ile266Thr | P | 0.03 | 0.003 | 7.40E-19 | 0.03 | 0.003 | 1.30E-23 | 0.03 | 0.002 | 4.47E-39 | 0.0 | 0.82 |
| rs3184504 | 12 | 111884608 | T/C | 0.483 | SH2B3 | Trp262Arg | P | -0.02 | 0.003 | 1.70E-11 | -0.02 | 0.003 | 1.10E-12 | -0.02 | 0.002 | 2.71E-22 | 0.0 | 0.95 |
| rs2229840 | 12 | 124826462 | T/C | 0.160 | NCOR2 | Thr1699Ala | P | 0.03 | 0.004 | 3.50E-34 | 0.03 | 0.003 | 7.70E-24 | 0.03 | 0.003 | 3.02E-40 | 0.0 | 0.97 |
| rs17068593 | 14 | 93118229 | T/C | 0.190 | RIN3 | Cys279Arg | P | 0.04 | 0.004 | 5.60E-34 | 0.04 | 0.003 | 1.50E-33 | 0.04 | 0.002 | 8.83E-62 | 0.0 | 0.43 |

**Table 1 (continued)**

| RSID | CHR | POS | EA/OA | FRQ | Gene | AA | Condition | Male (N = 205,513) | | | Female (N = 244,730) | | | Meta-analysis (N = 450,243) | | | | |
|---|---|---|---|---|---|---|---|---|---|---|---|---|---|---|---|---|---|---|
| | | | | | | | | BETA | SE | P | BETA | SE | P | BETA | SE | P | I²(%) | P$_{het}$ |
| rs35874463 | 15 | 67457698 | A/G | 0.942 | SMAD3 | Ile170 Val | S | −0.03 | 0.006 | 9.04E-06 | −0.03 | 0.005 | 6.96E-08 | −0.03 | 0.004 | 2.92E-12 | 0.0 | 0.84 |
| rs5742915 | 15 | 74336633 | T/C | 0.540 | PML | Phe645 Leu | P | −0.02 | 0.003 | 2.20E-18 | −0.03 | 0.003 | 3.20E-23 | −0.02 | 0.002 | 9.33E-39 | 0.0 | 0.71 |
| rs34949187 | 15 | 89386652 | A/G | 0.186 | ACAN | Gln275Arg | S | −0.04 | 0.004 | 9.65E-27 | −0.03 | 0.003 | 1.83E-16 | −0.03 | 0.002 | 4.75E-40 | 85.5 | 0.01 |
| rs116092985 | 16 | 2160973 | A/G | 0.904 | PKD1 | Trp1399Arg | P | 0.04 | 0.005 | 9.20E-17 | 0.04 | 0.004 | 2.70E-20 | 0.04 | 0.003 | 1.17E-34 | 0.0 | 0.80 |
| rs1136001 | 16 | 1531974 | T/G | 0.297 | NTAN1 | Asn283His | P | −0.02 | 0.003 | 9.10E-10 | −0.02 | 0.003 | 8.30E-09 | −0.02 | 0.002 | 3.02E-16 | 0.0 | 0.74 |
| rs9905106 | 17 | 1373518 | T/C | 0.265 | MYO1C | Gln826Arg | P | 0.01 | 0.003 | 6.30E-06 | 0.02 | 0.003 | 1.50E-09 | 0.02 | 0.002 | 7.51E-14 | 0.0 | 0.51 |
| rs34914463 | 17 | 7366619 | T/C | 0.869 | ZBTB4 | Asn561Ser | S | −0.04 | 0.004 | 4.58E-21 | −0.03 | 0.004 | 1.94E-20 | −0.04 | 0.003 | 9.81E-40 | 0.0 | 0.46 |
| rs36000545 | 17 | 79093822 | A/G | 0.604 | AATK | Phe1266Ser | P | 0.02 | 0.003 | 4.60E-13 | 0.02 | 0.003 | 4.40E-18 | 0.02 | 0.002 | 2.56E-29 | 0.0 | 0.52 |
| rs1786263 | 18 | 13116432 | T/G | 0.606 | CEP192 | Leu2449Arg | P | −0.02 | 0.003 | 4.80E-13 | −0.02 | 0.003 | 1.30E-13 | −0.02 | 0.002 | 1.03E-22 | 0.0 | 0.64 |

In the Condition column, "P" indicates an independent lead SNP identified before conditioning analysis, and "S" indicates a SNP identified in conditional analysis. RSID based on dbSNP, CHR chromosome, POS physical position based on the human genome assembly build 37 (GRCH37), EA effect allele, OA alternative allele, FRQ frequency of EA, AA amino acid change at the specified position, BETA beta coefficient of the linear mixed model, SE standard error of beta, P p value, I² statistics of SNPs in the meta-analysis; P$_{het}$ meta-analysis heterogeneity p value.

frequency mutations are more likely to play a direct role in changing protein function.

The top five loci containing missense lead variants are described below, while all 51 loci are listed in Table 1.

2p23.3 (*GCKR*). The common lead SNP rs1260326 (MAF = 39.6%, beta = −0.03, $p_{male} = 5.30 \times 10^{-29}$, $p_{female} = 4.20 \times 10^{-37}$, $p_{meta} = 6.16 \times 10^{-64}$) is located in the exon of the *GCKR* (glucokinase regulator) gene, resulting in an amino acid change from leucine to proline. This SNP was previously reported to be associated with multiple metabolic traits[26]. *GCKR* encodes a regulatory protein that inhibits glucokinase in liver and pancreatic islet cells by binding noncovalently to form an inactive complex with the enzyme.

14q32.12 (*RIN3*). The lead SNP rs117068593 (MAF 19.0%, beta = 0.04, $p_{male} = 5.60 \times 10^{-34}$, $p_{female} = 1.50 \times 10^{-33}$, $p_{meta} = 8.83 \times 10^{-62}$) results in a change from arginine to cysteine at the 204th amino acid of the protein encoded by the *RIN3* (ras and rab interactor 3) gene. This change is predicted to be harmful by PROVEAN, Polyphen-2 and SIFT. This locus was previously reported to be bivariately associated with both bone mass and lean mass at the lead SNP rs754388[18]. rs754388 is in nearly perfect LD with rs117068593 ($r^2 = 0.95$), implying the same association signal for both SNPs. The gene product of *RIN3* is a member of the RIN family of Ras interaction-interference proteins. It functions as a guanine nucleotide exchange for RAB5B and RAB31.

5q35.2 (*STC2*). The lead SNP rs148833559 (MAF 0.14%, beta = 0.38, $p_{male} = 6.70 \times 10^{-25}$, $p_{female} = 1.10 \times 10^{-30}$, $p_{meta} = 9.21 \times 10^{-52}$) is a rare mutation whose per allele effect is 10-fold larger than those of the above two common ones. The substitution of arginine to leucine at the 44th amino acid of the *STC2* (Stanniocalcin 2) protein is predicted to be harmful by PROVEAN, Polyphen-2 and SIFT. This rare variant was previously reported to be associated with human height[27]. In a recent study of ALM in middle-aged and older participants of the UKB cohort, the same SNP was identified. The authors also verified that knockdown of *STC2* had a significant effect on myotube length in C2C12 cells[15].

8q24.22 (*ZFAT*). Two missense SNPs in the *ZFAT* (zinc finger and AT-hook domain containing) gene are associated with ALM$_{adj}$. The primary SNP is a common SNP rs12541381 (MAF 25.8%, beta = −0.03, $p_{male} = 2.50 \times 10^{-19}$, $p_{female} = 3.30 \times 10^{-32}$, $p_{meta} = 2.81 \times 10^{-49}$), resulting in a change from proline to serine at the 102nd amino acid of the protein encoded by *ZFAT*, where the change is predicted to be benign by all four prediction tools. Conditional analysis identified a secondary rare SNP, rs112892337 (MAF 0.4%, beta = 0.14, $p_{male} = 4.53 \times 10^{-14}$, $p_{female} = 2.32 \times 10^{-11}$, $p_{meta} = 1.23 \times 10^{-23}$ after conditioning), which results in a change from serine to cysteine at the 470th amino acid. The change is predicted to be harmful by three prediction tools. *ZFAT* encodes a protein that likely binds DNA and functions as a transcriptional regulator involved in apoptosis and cell survival.

8q24.12 (*ENPP2*). The lead SNP rs10283100 (MAF 5.6%, beta = −0.06, $p_{male} = 4.00 \times 10^{-20}$, $p_{female} = 7.40 \times 10^{-29}$, $p_{meta} = 4.11 \times 10^{-44}$) results in a change from serine to proline at the 545th amino acid of the protein encoded by the *ENPP2* (Ectonucleotide pyrophosphatase/phosphodiesterase 2) gene. This change is predicted to be benign by all prediction tools. *ENPP2* encodes a protein functioning as both a phosphodiesterase and a phospholipase.

**Gene-based and geneset enrichment analyses**. A total of 2885 genes were significant at the gene-based GWS level ($\alpha = 0.05/17,788 = 2.81 \times 10^{-6}$, Supplementary Data 15). The most significant gene

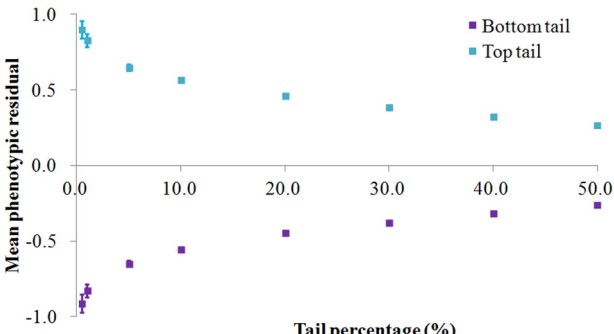

**Fig. 4 Polygenic risk score prediction.** A total of 277,504 participants were randomly selected as the training sample, and another independent 92,108 participants were selected as the validation sample. The variants achieving a $p$ value of $<1 \times 10^{-5}$ in the training sample were selected and used for prediction in the validation sample via the LDpred approach. Participants in the two extreme tails of the predicted genome-wide polygenic risk score (PRS) distribution were compared in terms of raw phenotypic mean (after correction). The $X$-axis represents the fraction of participants drawn from both extreme tails of the predicted PRS distribution. The $Y$-axis represents the mean $\text{ALM}_{\text{adj}}$ (with the standard error of the mean difference between the two tails).

was *DLEU1* ($p = 5.83 \times 10^{-170}$), followed by *ZBTB38* ($p = 1.94 \times 10^{-77}$), *FNDC3B* ($p = 9.68 \times 10^{-73}$), *EDEM2* ($p = 1.76 \times 10^{-72}$) and *CRADD* ($p = 2.36 \times 10^{-71}$). Using gene level association statistics as input, a total of 123 gene sets were significant at the geneset significance level ($\alpha = 0.05/15,481 = 3.23 \times 10^{-6}$, Supplementary Data 16). The top gene sets included genes with known functions related to the musculoskeletal and connective systems, such as GO:0001501 'skeletal system development' ($p = 3.31 \times 10^{-18}$, $N_{\text{gene}} = 483$), GO:0002062 'chondrocyte differentiation' ($p = 3.22 \times 10^{-14}$, $N_{\text{gene}} = 115$) and GO:0051216 'cartilage development' ($p = 3.55 \times 10^{-14}$, $N_{\text{gene}} = 197$).

**Polygenic risk score profiling**. To assess the ability of the GWAS findings to predict ALM, a polygenic risk score (PRS) analysis was performed on the subset of unrelated white participants from the UKB cohort. Three-quarters of the participants ($n = 277,762$, including 149,329 females) were randomly selected as the training sample, with the remaining participants ($n = 92,206$, including 49,660 females) as the validation sample.

The training sample revealed 134,277 variants with a $p$ value $<1 \times 10^{-5}$ for association with $\text{ALM}_{\text{adj}}$. Using these variants as predictors, the predicted genome-wide PRS and the real phenotype residual in the validation sample were significantly correlated (Pearson's correlation coefficient 0.32, 95% CI (0.32, 0.33), $p < 2.2 \times 10^{-16}$). Mean phenotype residuals in the top tail were significantly higher than those in the bottom tail of the PRS distribution (Fig. 4). For example, the predicted top 1% of participants had an increased average residual of 1.16 compared to that of the predicted bottom 1% of participants (0.57 (s.d 0.96) vs. −0.59 (s.d 0.94)), corresponding to a 1.69 kilogram (kg) increase in raw ALM (24.61 kg (s.d 5.89 kg) vs. 22.92 kg (s.d 5.27 kg)). In the female group, the predicted top 1% of participants had an average 1.39 kg increase in raw ALM compared with that of the predicted bottom 1% of participants (20.26 kg (s.d 2.75 kg) vs. 18.87 kg (s.d 2.45 kg)). In males, the increase was 2.29 kg (29.82 kg (s.d 4.18 kg) vs. 27.53 kg (s.d 3.56 kg)). These results demonstrate that PRS prediction based on the current GWAS finding is capable of identifying participants with high or low levels of ALM.

Using the same approach, we evaluated the capability of the GWAS finding to predict DEXA-derived ALM. The correlation

coefficient between PRS and DEXA-derived ALM was 0.18 (95% CI [0.15, 0.21] $p < 2.2 \times 10^{-16}$), again demonstrating the capacity of the present findings to predict DEXA-measured lean mass.

**Genetic correlations with other traits**. To test whether lean mass has a shared genetic aetiology with other diseases and relevant traits, a genetic correlation analysis was performed with LDSC[28]. $\text{ALM}_{\text{adj}}$ is strongly genetically correlated with whole body lean mass and ALM studied by a previous GWAS meta-analysis[14] ($r_g = 0.70$ and 0.56, $p < 2.2 \times 10^{-16}$) (Fig. 5). Strikingly, ALM was highly correlated with height ($r_g = 0.71$, $p < p < 2.2 \times 10^{-16}$), implying that the two traits share a large number of developmental pathways. Furthermore, $\text{ALM}_{\text{adj}}$ was modestly correlated with BMI ($r_g = 0.12$, $p = 1.59 \times 10^{-7}$). However, the correlation with heel bone mineral density was low ($r_g = -0.03$, $p = 0.13$). $\text{ALM}_{\text{adj}}$ was most negatively correlated with BMI-adjusted 2-h glucose ($r_g = -0.26$, $p = 7.79 \times 10^{-5}$) and BMI-adjusted leptin ($r_g = -0.23$, $p = 2.63 \times 10^{-7}$). It was also negatively correlated with body fat ($r_g = -0.21$, $p = 1.36 \times 10^{-10}$). However, this correlation should be interpreted with caution given the following two confounding factors. The first is collider bias, as the trait that we analysed was fat mass-adjusted lean mass. The second is the phenotypic constraint between lean mass and fat mass, as the sum of the two measures defines body weight and body size.

**Mendelian randomization analysis**. To investigate whether $\text{ALM}_{\text{adj}}$ is causally linked with other complex diseases and traits, a Mendelian randomization analysis was performed with GSMR[29]. Ten diseases and eight continuous traits from a variety of categories were chosen for evaluation. The scatter plot for all 18 traits is displayed in Supplemental Fig. 3. At the Bonferroni-corrected significance level of $2.78 \times 10^{-3}$ (0.05/18), $\text{ALM}_{\text{adj}}$ was causally associated with five diseases (coronary artery disease, $p = 2.09 \times 10^{-56}$; fracture, $p = 3.45 \times 10^{-10}$; type 2 diabetes, $p = 1.20 \times 10^{-8}$; insomnia, $p = 2.77 \times 10^{-5}$; and inflammatory bowel disease $p = 8.09 \times 10^{-4}$) (Supplementary Data 17). The causal association between $\text{ALM}_{\text{adj}}$ and type 2 diabetes was negative, indicating that $\text{ALM}_{\text{adj}}$ is a protective factor for the latter. A one-standard deviation increase in the $\text{ALM}_{\text{adj}}$ residual corresponded to a decrease in the odds ratio of 0.92 (95% CI [0.89, 0.95]).

At the same significance level, $\text{ALM}_{\text{adj}}$ was also causally associated with 6 metabolic traits, including four serum lipid traits and two blood pressure traits.

**Discussion**

This study of lean mass with approximately half a million participants, the largest sample used for a GWAS of lean mass so far, was successful. More than 1000 variants were identified at the genome-wide significance scale ($p < 5 \times 10^{-9}$). In particular, more than half of these variants achieved genome-wide significance ($p < 5 \times 10^{-9}$) in one sex and were replicated in the other sex ($p < 5 \times 10^{-5}$). Overall, these >1000 variants accounted for ~15% of ALM variation, again, the largest explainable fraction of variation in lean mass reported so far in a GWAS. Our finding of >1000 variants is expected for a complex trait with high heritability, particularly considering another trait with comparable heritability, height, for which ~700 variants were detected[30]. Interestingly, the majority of the loci detected in a previous smaller GWAS[12] and meta-analysis[14] of lean mass were also significant in the present study, providing solid evidence of replication.

The inability of GWAS to detect and replicate specific genetic variants for human complex traits, contradicting a trait's established high heritability, e.g. height, was formally recognized as the missing heritability problem a decade ago[31,32]. An explanation is the so-called polygenic model, where hundreds or even thousands

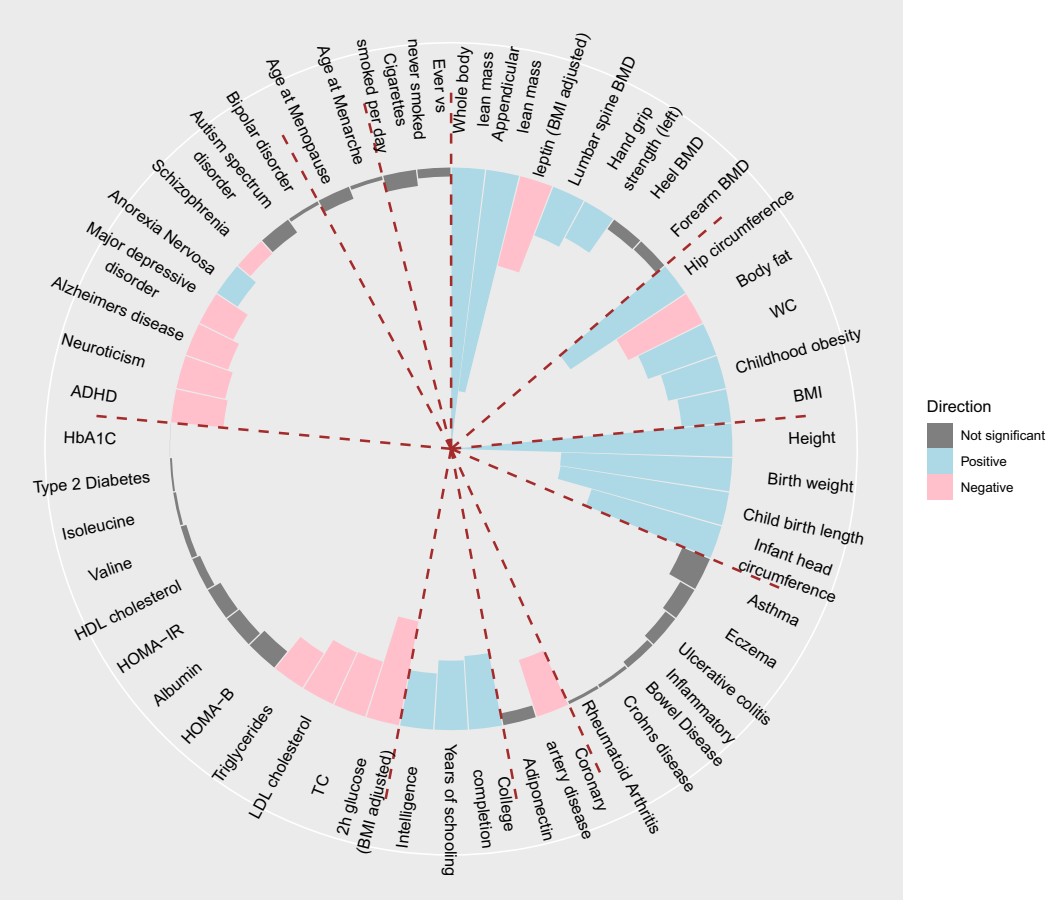

**Fig. 5 Genetic overlap with other traits.** Genetic correlations ($r_g$) between ALM$_{adj}$ and 51 traits and diseases were estimated. LD score regression tested for genome-wide SNP associations in these participants against similar data for various other traits and diseases, including musculoskeletal system, anthropometric, obesity, cognition, metabolism, psychiatry, reproduction and neuropsychiatric outcome traits. Error bars represent the standard errors of these estimates. Blue bars represent significant positive correlations at the nominal level of $p < 0.05$; pink bars represent significant negative correlations ($p < 0.05$); grey bars represent nonsignificant correlations.

of common SNP variants act additively, with each contributing only a tiny fraction of the trait variation. The genetic findings from the present study support this explanation for lean mass. The total 1059 conditionally independent variants explained 15.5% of the phenotypic variance, corresponding to an average per variant variance as low as 0.015%. It is worth noting that the present study had nearly 100% power to detect variants with an effect size larger than 0.015% and indeed did so. On the other hand, the power to detect variants with effect sizes as low as 0.001% was nearly zero. Therefore, there might exist more variants with effect sizes smaller than those identified in the present study, further supporting the polygenic model.

The functional relevance of our identified variants was supported by geneset enrichment analysis, where GO terms, including GO:0001501 'skeletal system development', GO:0061448 'connective tissue development' and GO:0051216 'cartilage development', were among the significant gene sets. Specifically, the common genes involved in these terms were tightly connected into a network that contained TGF pathway genes, BMP pathway genes and SMAD family genes, which are all important musculoskeletal development genes/pathways. This finding is concordant with knowledge of developmental biology since cells from bone, cartilage, muscle and fat share the same progenitor, mesenchymal stem cells, and pleiotropy of muscle and bone is well recognized in both humans[33] and animal models[34].

To declare an association as significant, we required that the signal not only be significant at the GWS level in the combined analysis but also be significant at the $5 \times 10^{-5}$ level for each sex group. This significance level was essentially equivalent to that in a two-stage design, where the first stage involved a GWAS in one group (e.g. the male group) and the second stage involved replicating top hits in the other group (e.g. the female group). As a maximal number of 1000 independent loci was assumed, we could have selected the top 1000 hits from the first stage for replication at the second stage. As a result, a significance level of $5 \times 10^{-5}$ (0.05/1000) was sufficiently conservative to declare successful replication. In our actual analysis, the numbers of independent loci with $p < 5 \times 10^{-5}$ were 1988 and 1713 in the female and male groups, respectively, which were almost twice the presumed number ($n = 1000$) of independent loci. This may have inflated the type I error rate for the variants whose $p$ values fell within the range from $5 \times 10^{-5}$ to $5 \times 10^{-9}$ (i.e. the Tier 3 variants).

The present study had the following strengths. First, the large sample size of over 400,000 participants is the largest used for a lean mass GWAS to date, offering a unique opportunity to discover loci that were undetected by previous smaller GWASs. Second, instead of analysing the sample as a whole, the two sexes were analysed separately, and then meta-analysis was performed. This may have reduced the statistical power for identifying new loci but allowed us to replicate significant findings between the sexes. Third, via a series of comprehensive downstream analyses annotating the identified SNPs, a deep understanding was achieved for the genetic mechanism underlying ALM and its interplay with other complex traits and diseases.

Certain limitations existed in the present study. First, lean mass was measured by the BIA approach, which is not as reliable as the gold standards for quantifying lean mass, such as magnetic resonance imaging and computed tomography, because these latter methods are direct measures. Instead of measuring lean mass directly, BIA derives an estimate of lean mass based on electrical conductivity. Therefore, it can be influenced by the hydration status of the subject. Moreover, the derivation equation of BIA relies on a calibrated reference population, which may not be well validated across populations. Second, the genetic findings for lean mass alone are inadequate for characterizing the full genetic basis of sarcopenia. This is because there is a consensus that sarcopenia is defined not only by low lean mass but also, more importantly, by low muscle strength and poor physical performance[35]. Therefore, the present study only discovered the genetic mechanism of sarcopenia from the lean mass perspective, far more than enough to begin understanding the genetic basis of sarcopenia as a whole. Third, physical activity is known to influence lean mass[36], the confounding effect of which was not controlled for in the present study.

In summary, we performed a GWAS using approximately a half-million participants for lean mass. The variation (~15%) in lean mass explained by the identified variants represents a significant leap in explaining the hidden heritability of this complex trait using the GWAS approach. The translational value of these findings lies in the importance of lean mass for other complex diseases, such as type 2 diabetes, as our Mendelian randomization analysis showed that ALM is a protective factor for the latter. Overall, our study provides another example in which a GWAS with a very large sample size ultimately and thoroughly delineates the genetic architecture of a complex human trait. This epitomizes the value of big data in human genetic research.

## Methods

**Study participants**. The study sample came from the UKB cohort, which is a large prospective cohort of ~500,000 participants from across the United Kingdom, aged between 48 and 73 at recruitment. Ethics approval for the UKB study was obtained from the North West Centre for Research Ethics Committee (11/NW/0382), and informed consent was provided by all participants. This study (project number 41542) was covered by general ethical approval for the UKB study.

As careful data quality control is critical to avoid false positives, we only analysed self-reported white individuals (data field 21000). Participants who had a self-reported sex inconsistent with the genetic sex, whose sex chromosome was aneuploid, who had unusually high heterozygosity and/or missing rates or who withdrew their consent were removed. Overall, 487,378 participants were identified to have both phenotypic and genotypic data, 37,135 of whom were excluded. The final sample consisted of 450,243 participants, including 244,730 females and 205,513 males.

**Phenotype and modelling**. Body composition was measured by the BIA approach. ALM was quantified by the sum of fat-free mass at the arms (data fields 23121 and 23125) and legs (data fields 23113 and 23117). Appendicular fat mass (AFM) was quantified by the sum of fat mass at the arms (data fields 23120 and 23124) and legs (data fields 23112 and 23116). In each sex, covariates, including AFM, age, age squared, the top 10 principal components, assessment centre (23 levels) and genotyping array (2 levels), were used to adjust raw ALM values. To avoid collider bias due to height, height was not included as a covariate. The adjusted residuals were normalized into inverse quantiles of a standard normal distribution, which were used for subsequent association analysis.

A small subset of 4294 participants also received a DEXA body composition scan, and hence, their DEXA-derived ALM was also available. Therefore, raw ALM values derived from DEXA and from BIA were compared in these participants to evaluate phenotypic consistency from these two measurements.

**Genotype quality control**. Genome-wide genotypes were available for all participants at 784,256 genotyped autosome markers and were imputed into UK10K haplotype, 1000 Genomes project phase 3 and Haplotype Reference Consortium reference panels. Genotype imputation was performed by the UKB. A total of ~92 million variants were generated by imputation. We excluded variants with a MAF < 0.1% and with an imputation $r^2 < 0.3$. As a result, ~19.4 million well-imputed variants were retained for subsequent analysis.

**Genetic association analysis**. In each sex group, we used BOLT-LMM to perform linear mixed model analysis[17]. Upon the completion of sex-specific association analyses, we meta-analysed the summary statistics of the two sexes by an inverse-variance-weighted fixed-effects model with METAL[37]. The GWS level was set at α = $5 \times 10^{-9}$ to account for both common and rare variants[38]. The variants that passed this threshold in the between-sex meta-analysis were then checked for significance in both sexes. Previous studies on human height and body mass index (BMI), two of the most polygenic traits, identified 712 and 536 loci in up to ~700,000 European participants[39]. By referring to these previous findings, we selected an arbitrary but reasonable number, i.e. 1000, for the maximal number of independent loci for ALM$_{adj}$ and set the suggestive significance level to be $5 \times 10^{-5}$ (0.05/1000) to account for multiple testing. An association was declared only if the signal was significant at the GWS level ($p < 5 \times 10^{-9}$) in the meta-analysis and was significant at the suggestive level ($p < 5 \times 10^{-5}$) in both sexes.

Differences in genetic effects between females and males were examined by a two-sided z-score test with the following equation:

$$z = \frac{\beta_{female} - \beta_{male}}{\sqrt{var(\beta_{female}) + var(\beta_{male})}}$$

where $\beta_{female}$ and $\beta_{male}$ are regression coefficients for females and males and var(·) are their variances, respectively.

The identified variants were annotated by Variant Effect Predictor (VEP)[40], which invokes dbNSFP[41] to annotate nonsynonymous SNPs.

**Conditional association analysis**. To identify additional signals in regions of association, approximate joint and conditional association analysis was performed using the GCTA tool[42]. From the UKB sample, a reference sample of 100,000 unrelated participants was generated for estimating LD patterns. Specifically, KING software[43] (with a kinship coefficient cut-off at 0.0884) was used to infer 369,968 unrelated participants, from whom the 100,000 participants of the reference sample were randomly drawn.

A recursive conditional association analysis was performed. In each iteration, an approximate conditional analysis conditioned on the current list of lead variants was performed in each sex, followed by a between-sex meta-analysis. Again, an association was defined as significant if it achieved both a conditional meta-analysis GWS signal and a conditional suggestive signal ($p < 5 \times 10^{-5}$) in both sexes. In addition, each such identified variant was required to be independent of all variants in the lead SNP list (LD $r^2 < 0.1$). The variant with the lowest $p$ value among such identified variants was added to the list of lead variants. Iterations of the conditional analysis were run until no significant signal could be detected.

**Overlap with loci related to obesity traits**. GWAS summary statistics for three obesity traits, including BMI[39], waist circumference and waist-hip ratio[44], were downloaded from the GIANT consortium website. For each trait, SNPs located within all the identified loci (lead SNP +500 kb flanking region to either side) were extracted from the GWAS summary statistics. The significance level for the obesity traits was set at the conventional level of $5.0 \times 10^{-8}$.

**Replication in the South Asian population**. The UKB participants from the South Asian population were analysed to replicate the findings identified in the white population. Specifically, self-reported South Asian participants (data field 21000, including Indian, Pakistani and Bangladeshi) were collected. Quality control criteria were the same as those for the main analysis. The final sample consisted of 7452 participants. Phenotype modelling was the same as that for the main analysis, with the only exception that both sexes were analysed together, and sex was used as a covariate. Association analysis was performed again with BOLT-LMM.

**Comparing imputation accuracy with exome-sequencing data**. During the preparation of this manuscript, the UKB released exome-sequencing data on a selected subset of ~50,000 participants. While we are aware of the systematic assembly issue announced by the UKB, we used the current released SPB dataset to assess the imputation accuracy of the present GWAS data by comparing association signals from imputed genotypes to those from direct sequencing. BOLT-LMM involves sophisticated two-step implementation, where the first step fits the model to a set of directly typed variants and the second step examines the association in another set of imputed and/or typed variants. When using BOLT-LMM for comparison, the first-step model parameters fitted to imputed versus sequenced data may incur differences, which is nonrelevant to imputation accuracy. To make the comparison as fair as possible, we used a simple linear model instead. Specifically, we generated an unrelated sample consisting of participants who were both exome-sequenced and genotype-imputed.

As the QC procedure, we removed participants who were not self-reported as white, whose self-reported sex was inconsistent with their genetic sex, and who withdrew their consent. KING software was again used to select unrelated participants[43] according to genotyped SNPs. The final sample consisted of 45,554 participants, including 24,740 females and 20,814 males.

Sequence variant coordinates, which were annotated to the GRCH38 assembly, were converted back to the GRCH37 assembly with Liftover (http://genome.ucsc.

edu/cgi-bin/hgLiftOver). For each participant, variants that were missing in the sequenced data were set to missing in the imputed data as well. In both datasets, genetic association with normalized phenotype residuals was analysed in *R*.

**Genetic architecture**. The LDSC method was used to estimate the amount of genomic inflation due to confounding factors such as population stratification and cryptic relatedness[28]. Precomputed LD scores from the 1000 Genomes Project for European participants were used for estimation. The relative contribution of confounding factors was measured by the attenuation ratio, which is defined as (intercept-1)/(mean chi^2−1), where intercept and mean chi^2 are estimates of confounding and the overall association inflation, respectively[28].

BOLT-REML was used to estimate heritability tagged by all the analysed variants[19]. It was applied to raw genotypic data with or without removing all variants among the total identified loci (each defined as the lead SNP +500 kb flanking region to either side) to estimate respective values of heritability. The difference between the two measures provides an estimate of the heritability explained by the identified loci.

The variance explained by all lead variants was calculated as the sum of all individual variant effect sizes, which is defined as the percentage of phenotypic variance explained by the variant and estimated with the formula $2f(1-f)\beta^2$, where $f$ is allele frequency and $\beta$ is the regression coefficient associated with the variant. To account for the winner's curse effect, the effect sizes were shrunk with a false discovery rate (FDR)-based method[45].

**Age-dependent effect**. The identified lead variants were evaluated for their age-dependent effects on $ALM_{adj}$. Specifically, the sample was divided into the following six age strata defined by bins of 5 years: 45 or less ($N = 54,608$), 46–50 ($N = 58,865$), 51–55 ($N = 70,253$), 56–60 ($N = 89,479$), 61–65 ($N = 109,696$) and 66 or more ($N = 67,342$). Participants of both sexes within each age stratum were pooled together for analysis. Phenotype modelling within each age stratum was similar to that in the main association analysis, with the only exception that sex was added to the covariates. The association within each age stratum was examined by BOLT-LMM. The generated regression coefficients from all age strata were meta-regressed against the mean age of each stratum to examine the effect of age on the genetic effect. Meta-regression was implemented by a linear regression analysis weighted by the inverse variance of each regression coefficient. Evidence of significance was determined by the Bonferroni-corrected significance level.

**Enrichment analysis**. Stratified LDSC was used to partition heritability from GWAS summary statistics into different functional categories[20]. The analysis was based on the 'full baseline model' created by Finucane et al.[20] from 24 publicly available main annotations that are not specific to any cell type. The significance level of enrichment was set at $p < 2.08 \times 10^{-3}$ (0.05/24).

Stratified LDSC was also used to assess the enrichment of heritability in specific tissues and cell types[21]. This method analyses gene expression data together with GWAS summary statistics, for which the two precompiled gene expression datasets in LDSC were used. The first is the GTEx project dataset[46], and the second is the Franke Laboratory dataset[47]. The GTEx dataset contains 53 tissues with an average of 161 samples per tissue. The Franke Laboratory dataset is an aggregation of publicly available microarray gene expression datasets comprising 37,427 human samples from 152 tissues. A total of 205 (=53 + 152) tissues were classified into nine categories for visualization. Again, evidence of significance was determined by a Bonferroni-corrected significance level of $p < 2.44 \times 10^{-4}$ (0.05/205).

**Candidate gene prioritization**. At each associated locus, CRVs were defined as variants in strong LD with the lead variant ($r^2 > 0.8$, including the lead variant itself). The LD $r^2$ measure was estimated based on the above 100,000 unrelated reference samples with PLINK[48]. Six sources of information were used to evaluate a gene's causality: (1) being nearest to the lead CRV; (2) containing a missense coding CRV; (3) being a target gene for a cis-eQTL CRV; (4) being a target gene for a cis-protein QTL (cis-pQTL) CRV; (5) being prioritized by DEPICT analysis[49] and (6) being prioritized by SMR analysis[50].

Cis-eQTLs revealed by the GTEx (v7) project were accessed from the GTEx web portal (www.gtexportal.org/)[46]. Cis-eQTL information is available for over 50 tissues. We selected skeletal muscle and whole blood for our analysis. Cis-eQTL was searched within a 500 kb distance of the target gene. Significant cis-eQTLs were declared at $p < 5 \times 10^{-5}$.

Cis-pQTL information was accessed from Sun et al.[51]. GWAS summary statistics for 3284 proteins were downloaded from the study's website. Cis-pQTL was searched within a 500 kb distance of the target gene. Significant cis-eQTLs were declared at $p < 5 \times 10^{-5}$.

DEPICT is an integrative tool that takes advantage of predicted gene functions to systematically prioritize the most likely causal genes at loci of interest[49]. The input of DEPICT includes a list of variant identifiers, and the output contains all genes located in the loci and their $p$ values as candidate genes. All lead variants were submitted to DEPICT for analysis. Significant genes were declared at a false discovery rate <5%.

The SMR (Summary data–based Mendelian Randomization) method[50] is another gene prioritization program that integrates summary-level data from

GWASs with data from eQTL studies to identify genes whose expression levels are associated with traits due to causal or pleiotropic effects. Here, the pleiotropy effect means that a SNP is causally associated with both gene expression and phenotypic variation. SMR uses SNPs as an instrumental variable and tests the causal relation of gene expression to phenotype variation. The results are interpreted as the effect of gene expression on the phenotype free of confounding from nongenetic factors. We used a precompiled eQTL dataset of whole blood tissue for estimation[52]. Evidence of a pleiotropic instead of causal relationship between an eQTL and $ALM_{adj}$ was examined by the HEIDI test[29]. We set a loose significance level of 0.05 for the HEIDI test to exclude potential pleiotropy.

The intersections of candidate genes prioritized from different sources were plotted using the R package UpSetR[53].

**Gene-based and geneset enrichment analyses**. Gene-based association analysis was performed with MAGMA v1.6[54], as implemented on the FUMA website (http://fuma.ctglab.nl/). GWAS meta-analysis summary statistics were mapped to 19,427 protein-coding genes, resulting in 17,788 genes that were covered by at least one SNP. A gene-based association test was performed, taking into account the LD between variants. The significance level was set at a stringent Bonferroni-corrected threshold of $2.81 \times 10^{-6}$, i.e. 0.05/17,788.

The generated gene-based summary statistics were further used to test for enrichment of associations with specific biological pathways or gene sets. A geneset's association signal was evaluated by integrating all signals from the genes in the set with MAGMA. A competitive geneset analysis model was used to test whether the genes in a geneset were more strongly associated with the phenotype than other genes.

Gene sets were obtained through the MSigDB website (http://software.broadinstitute.org/gsea/msigdb/index.jsp)[55]. Each gene was assigned to a geneset as annotated by Gene Ontology (GO), Kyoto Encyclopedia of Genes and Genomes (KEGG), Reactome and BioCarta geneset databases and other gene sets curated by domain experts or from the biomedical literature[55]. A total of 15,481 gene sets were used in this analysis. Significance was set at a Bonferroni-corrected level of 0.05/15,481 = $3.23 \times 10^{-6}$.

**Polygenic risk score profiling**. To assess the capability of the GWAS findings to predict ALM, a PRS analysis was conducted with a training sample and a validation sample. It was required that the two samples be independent from each other. To accomplish this, both samples were drawn from unrelated participants extracted from the main UKB sample. Specifically, three-quarters of the participants (277,504, including 149,172 females) were randomly selected as the training sample, and the remaining one quarter of the participants (92,108, including 49,604 females) were selected as the validation sample. Participants of both sexes were pooled together for analysis.

Raw phenotypes were adjusted by age, age squared, sex, assessment centre, genotyping array, AFM and the top 10 PCs. The residuals were converted to the standard normal distribution quantiles for downstream analysis. Genetic association analysis was performed with PLINK 2[56] because of its computational efficiency.

PRS calculation was conducted with LDpred[57]. LDpred infers the posterior mean effect size of each marker by using a prior of effect sizes and LD information from an external reference panel. To save computer memory usage, only the variants achieving a $p$ value of $<1 \times 10^{-5}$ in the training sample were selected and used for prediction in the validation sample. Specifically, the validation sample with original genotypes was used as a reference panel for LD estimation. The number of SNPs used to adjust LD from each side of the target SNP was set to 1000. Other software parameters were set to the default.

Using the same approach, we also evaluated the capability of the GWAS finding to predict DEXA-derived ALM. To accomplish this, we divided the total 450,243 eligible participants into one training sample including participants who did not receive the DEXA scan and one validation sample including participants who received the DEXA scan. The correlation between the risk score and DEXA-derived ALM was examined in the validation sample.

**Genetic correlations with other traits**. To test whether lean mass has a shared genetic aetiology with other diseases and relevant traits, a genetic correlation analysis was performed with the LDSC method[28]. An online web tool, LDHub (http://ldsc.broadinstitute.org/ldhub/), was used to estimate the genetic correlations between $ALM_{adj}$ and 49 complex traits and diseases. The standalone version of the software was used to estimate the correlations between $ALM_{adj}$ and two additional traits, ALM and whole body lean mass, which are not available in the LDHub GWAS summary statistics collections and were downloaded from the GEFOS consortium website (http://www.gefos.org).

Both the LDHub and standalone analyses adopted the same QC criteria. Specifically, only HapMap3 autosomal SNPs were included to minimize poor imputation quality[28]. SNPs were further removed given the following conditions: MAF < 0.01, palindromic strand (A/T or C/G), duplicated ID, or reported sample size less than 60% of the total sample size. LD scores precomputed with the European participants in 1000 Genomes Project were used for calculation.

**Mendelian randomization analysis**. To investigate whether $ALM_{adj}$ (as exposure) is causally associated with complex traits and diseases (as outcomes), a Mendelian randomization analysis was performed with GSMR[29] on 10 diseases and 8 metabolic traits. The 10 diseases included fracture, type 2 diabetes, asthma, insomnia, inflammatory bowel disease, smoking addiction, coronary artery disease, amyotrophic lateral sclerosis, bipolar disorder and autistic spectrum disorder. Although some of the selected diseases were distantly related to lean mass, they could serve as negative controls for the MR analysis. The metabolic traits included high-density lipoprotein cholesterol, low-density lipoprotein cholesterol, total cholesterol, triglycerides, insulin, glucose, diastolic blood pressure and systolic blood pressure.

GWAS summary statistics for each trait were downloaded from the corresponding study's website. From the list of SNPs associated with $ALM_{adj}$ at the $5 \times 10^{-9}$ level, qualified SNPs were included based on the following criteria: concordant alleles between exposure and outcome GWAS summary statistics, nonpalindromic SNPs with certain strands, a MAF > 1%, and allele frequency difference between exposure and outcome GWAS summary statistics <0.2.

Independent SNPs were further clumped with PLINK 2[56] by using an independence LD threshold $r^2 < 0.05$ and a 1 MB window size. For each pair of studied traits, the clumped independent SNPs were examined for their pleiotropic effects on both exposure and outcome by the HEIDI test[29]. The significance level for the HEIDI test was set to $\alpha = 0.05$. After removing pleiotropic SNPs on an outcome-by-outcome basis, the remaining independent SNPs were taken as instrumental variables to test for a causal effect of exposure on outcomes. The estimated causal effect coefficients are approximately equal to the natural log odds ratio for a case–control trait. The MR analysis significance level was set to $2.78 \times 10^{-3}$ (0.05/18).

**Reporting summary**. Further information on research design is available in the Nature Research Reporting Summary linked to this article.

## Data availability

Genome-wide summary statistics are available through the NHGRI-EBI GWAS Catalog (https://www.ebi.ac.uk/gwas/downloads/summary-statistics), with accession IDs GCST90000025-GCST90000027. Additional data are available in the Supplementary Data file in Supplementary Data 1–17

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

## Acknowledgements

This research was conducted using the UK Biobank resource under application number 41542. Y.F.P. and L.Z. were partially supported by funding from the National Natural Science Foundation of China (31771417 and 31571291), and by a project funded by the Priority Academic Program Development (PAPD) of Jiangsu Higher Education Institutions. Y.Z.L. was supported by GoMRI (Gulf of Mexico Research Initiative) grant G-23818 and NIH grants NIH/Fogarty 1U2RTW010673-01, R01 MH116844-01 and R01 AI147372-01. The numerical calculations in this paper were performed on the supercomputing system of the National Supercomputing Center in Changsha.

## Author contributions

Y.F.P. and L.Z. designed and supervised the study. L.Z. collected the data. Y.F.P., L.Z., X.L.Y. and H.Z. analysed the data. G.J.F. and X.T.W. performed the literature review and provided technical assistance. Y.F.P. and Y.Z.L. interpreted the data and drafted the early version of the manuscript.

## Competing interests

The authors declare no competing interests.
