## [Peer Review File · Communications Biology]

Reviewers' comments:

Reviewer #1 (Remarks to the Author):

The authors present a comprehensive genetic architecture study of appendicular lean mass (ALM) using UK Biobank data. This study is valuable to understand the genetic basis of ALM since it is the largest sample size used to date. My comments are basically based on suggestions for a better comprehension and try to enrich the study. The manuscript is well written, easy to follow and the methods are very well explained in detail. My specific comments are provided below:

Major comments:

1. Study participants. Why the authors select self-reported white individuals? The authors carried out the genetic association analysis using linear mixed models to control for population structure. Furthermore, it is not clear for me why the authors discarded individuals genotyped but not imputed. In total, how many individuals are removed from the initial dataset?
2. Candidate genes prioritization. Is the *PAM* gene the only one that has the six sources of evidence? What about the combinations of five, four, three, etc. source of evidences? It could be good if the authors could deep into the results by exploring the sources using for example the UpSetR package (<https://cran.rstudio.com/web/packages/UpSetR/vignettes/attribute.plots.html>)
3. Mendelian randomization analysis. How many variants were used for the MR analysis for each disease/metabolic trait? It could be nice if the authors could make a Figure to visualize the results, as for example, a scatterplot of the betas and the regression line analyzed for both ALM and each disease/metabolic trait.
4. In order to enrich the whole analysis performed in this valuable work the authors could make a flowchart (detailing number of individuals, significant variants, variants used in post GWAS-analysis, etc..) including GWAS for each gender, meta-analysis, conditional analysis, genetic architecture, candidate genes, mendelian randomization analysis, etc.

Minor comments

- 1- Line 102. Genotype quality control. How do you phase haplotypes of the UK Biobank individuals? How do you impute variants? You explain that you use three reference panels, what do you do in case that a variant is present in more than one panel? Are the ~19.4 million variants unique?
- 2- Lines 141 and 172. KING software. What threshold of the kinship coefficient do you use to select unrelated individuals in these analyses?
- 3- Line 156. "561 identified loci", these loci are identified from the genetic association analysis (Line 109) and using $P < 5e-9$?
- 4-Line 303. I do not find the links to download the GWAS summary statistics.
- 5-Line 336. "controlling for AFM". Do you use AFM as covariate in the GWAS analysis? If yes, I would add this in Lines 109-114. I understood that AFM was adjusted only in the transformation of ALM as explained in the methods section (Line 88).
- 6-Lines 371-375. Are these results obtained by comparing with the GIANT Consortium? If yes, please, clarify this.

8- Lines 423-424. "PAM is also prioritized by both SMR and DEPICT.." This statement is redundant since at the beginning of the paragraph is mentioned that PAM has the six sources of evidence.

9- Line 445. "Majority of these 42 variants". Please, quantify with the number or percentage of variants.

10- Typo in the title of the submitted manuscript. "Appendilar".

Reviewer #2 (Remarks to the Author):

In this study, Pei et al. perform a GWAS of appendicular lean mass in ~450,000 participants of the UK Biobank (UKB) of European ancestry. The authors detect 717 variants associated with ALM. They then assess the predictive accuracy of a resulting polygenic score and perform a Mendelian randomisation to test if variation in ALM are causal of certain diseases like type 2 diabetes. Overall, I found the paper well written and clear in its message. I noted below a few comments, which, if addressed, would improve the quality of the manuscript.

Major comments:

Ancestry. As far I can judge, they authors have determined ancestry based on self-report, which may be unreliable in many instances. In this cases, method like mixed-models GWAS implemented in BOLT may be unsuccessful to fully correct for stratification. I'd recommend that the authors, at least use projected PCs to call ancestry. I think this has been done centrally by the UKB so the authors could request that information.

Replication. The authors propose to assess the replication or replicability of their results by looking up the association within sexes. I don't think this is a proper replication approach and therefore I'd recommend that the authors remove it. They could for example either perform a look up in summary statistics from previous GWAS or they could look up in non-European ancestry participants the UKB. In particular within South-Asian ancestry participant who show the smallest F_{st} (differentiation) with European ancestry participants.

Collider bias. Many of the genes/loci highlighted by the authors seems to overlap with those of height. I'm wondering to which extent this could be due to collider biases given that the authors have adjusted their analyses for height (and height squared), which is an heritable trait. Can the author perform an analysis unadjusted for height and show correlation of effect sizes? Also, check how many of these 717 SNPs remain genome-wide significant when not adjusting for height?

Variance explained. The authors calculate the variance explained by the genome-wide significant SNPs by simply summing up the $2p(1-p)b^2$, where p is the minor allele frequency (MAF) and b is the estimated SNP effect. Because of estimation error (although small here given and sample size is quite large) and most importantly because of winner's curse this estimator of the variance explained is necessarily inflated. The ideal case would be to re-estimate the variance explained in an independent sample. Or at the very least perform a correction for Winner's curse.

MR. The authors applied a test for heterogeneity (HEIDI) to exclude the possibility that their finding are driven by pleiotropy and not by causality. The threshold chosen ($p < 1e-5$; line 311) is too lenient for HEIDI. People use 0.05 if you want to be sure that your results are not driven by pleiotropy. It would suggest this threshold instead.

Minor comments.

Line 58: Despite the large and NOT Despite of

Line 59: percentage of

Line 71: specify here if the >700 SNPs are independent

Line 73: please remove "important" here

Line 93: Put a reference here or point to the method section with more details

Line 117: Put a reference there to justify the threshold (e.g.

<https://www.ncbi.nlm.nih.gov/pmc/articles/PMC6351539/> or

<https://genomebiology.biomedcentral.com/articles/10.1186/s13059-017-1216-0>)

Line 122: why guess a reasonable number when you can directly assess how many loci you give based on a certain definition (e.g. that of Wood et al 2014)

Line 124-126: This is not a replication per se.

Line 141: UKB team should have provided groups with a list of related individuals. Have you used that list or did you rerun KING to identify them?

Line 156: Not quite sure where this number comes from?

Line 172: Could you please specify here that you used KING with genotyped data and no imputed?

Line 190: Are these 369,968 identified using KING? Please clarify here.

Line 231: SMR is not a SNP prioritisation method but a GENE prioritisation method.

Line 270: I think that "computational" would sound better here than "running".

Line 273-274: That looks like a trick to make LDpred run faster, right? Please clarify and justify why you've pre-selected these SNPs. In principle LDpred could run without.

Line 303: Please rephrase as the sentence does not make sense.

Line 351: Did you use clumping to identify these loci? Please clarify.

Line 371: Are these 302 SNPs the most associated one vs the 259? The conclusion is a bit rushed otherwise.

Line 381-388: Your set of SNPs is ascertained to have limited differences between sexes. Therefore you lack power to identify sex-specific effects. That should be acknowledged somewhere (discussion probably).

Line 387: please report these effects here.

Line 393: "As expected..." please justify why this relation expected? Do we have evidence of negative selection for this trait?

Line 400: ER (enrichment ratio) would be better than EA.

Line 421: rs400596 is an eQTL for PAM in blood. Do you also have the strength of association with expression of PAM in skeletal muscle? Please report it as that would add to your demonstration.

Line 424: its potential function relevance

Line 433: Please replace "are not able to derive" with "could not be calculated". Do you know why? Is it because there were not many I suppose.

Line 438-443: $r^2 > 0.3$ is too lenient for rare variants. The main challenge with imputation is allele

frequency. So Are the 33 variants common or rare? If the former then, this is expected.

Line 556: Your analyses were adjusted for AFM, then you find a negative correlation with fat mass. This looks like collider to me.

Line 593: single variant do not have heritability. You could instead speak about variance explained by the variant.

Line 797: Could you please specify what curve has been fitted?

Reviewer #3 (Remarks to the Author):

Pei et al explored the genetic basis of human appendicular lean mass in the UK Biobank, representing the largest to date sample for this trait. The paper is easy to follow, the analysis is well-done, the applied methods are appropriate and the follow-up analyses are rich and interesting. I only have relatively minor issues, which I list below.

Most importantly, the work and the findings need to be put in context of the available literature. An important piece (ALM GWAS in a younger subset of the UK Biobank excluding age related muscle loss) is missing: <https://www.ncbi.nlm.nih.gov/pubmed/31761296> Thus, it is important to carefully relate the findings to those published earlier.

Major points

1. In relation to the publication mentioned above, it would be interesting to see age-specific effects (how do the effect of the discovered SNPs compare in middle-aged vs elderly?).
2. Are the authors aware of the issues with the currently released versions of the exome sequencing data from the UK Biobank [systematic under-marking of duplicate reads in the SBP pipeline and under-calling of variants in primary assembly regions from which the alternative contigs are derived for the FE exome dataset.]? How was this handled?
3. "Genes involved in multiple gene sets are likely to act as hub genes and may play a central regulatory role." While this would be true in an ideal world, the way GO and other gene sets are defined are highly bias towards more studied genes, generating a vicious circle. Also the fact that the 2nd and 3rd most often appearing gene show no significant gene-level score, supporting no link with the disease, may simply represent study-bias.
4. "It is also negatively correlated with body fat ($r_g = -0.17, 9.44E-6$), suggesting an opposite developmental direction towards lean and fat mass." – I'm not sure I agree with the interpretation. Any SNP that is responsible for general body size (height or weight) will show an opposite effect on lean and fat mass, because it is associated with something that is related to the sum of these two measures.
5. As a proof of concept and support of the bioimpedance based measure, could the derived PRS be correlated with the DXA-measured lean mass (in up to 5K samples)? A score that explains 10% of trait variance must have an easily detectable correlation in 4k samples.
6. I wonder whether the authors see any relationship between (self-reported or preferably actimeter-measured) physical activity and lean mass? If so, should it be corrected for?
7. While the authors corrected for height and height², I was wondering how similar their trait residuals are to a lean body mass index (=lean mass/(height²)) measure?

Minor points

1. "Phenotypic outliers were monitored by the Tukey method." – You should be more specific, have those values been removed? Did you use $k=1.5$?

2. Excluding imputation $r^2 < 0.3$ is not stringent enough, usually people keep SNVs with $\hat{r}^2 > 0.7$ or 0.8 How many of your top hits fall in the $0.3 < r^2 < 0.7$ category?
3. "across-gender meta-analysis" -> "between-gender meta-analysis"
4. "almost all of the identified variants may have similar effect sizes across genders" - It has to be pointed out that this is mainly due to the selection of the 717 SNPs. Those are significant in sex-combined analysis, thus expected to have similar effects between sexes. The way to detect sex-specific associations is to go deeper in sex-combined association P-values, where one has a chance to detect effects specific to one sex only, for example. I understand that sex-specificity is out of the scope of the paper, but the statement needs to be adjusted not to mislead the readers.
5. "Though the 7 rare variants are GWS significant in the whole cohort, they appear to have limited imputed dosage variation in the much smaller exome-sequenced sample...", still the concordance between imputed and sequenced genotypes can be established. What is the r^2 and D-prime for these 7 SNVs?
6. Please have the manuscript checked by a native English speaker to avoid sentences like "hence their imputation association p-values are not able to derive"
7. "3,101 genes are significant at the gene-based GWS level" - This gives very little insight into biology. As sample sizes increase, all genes will become significant (e.g. height GWAS paper in preparation). The importance is their relative order.

In this study, Pei et al. perform a GWAS of appendicular lean mass in ~450,000 participants of the UK Biobank (UKB) of European ancestry. The authors detect 717 variants associated with ALM. They then assess the predictive accuracy of a resulting polygenic score and perform a Mendelian randomisation to test if variation in ALM are causal of certain diseases like type 2 diabetes. Overall, I found the paper well written and clear in its message. I noted below a few comments, which, if addressed, would improve the quality of the manuscript.

Major comments:

Ancestry. As far I can judge, they authors have determined ancestry based on self-report, which may be unreliable in many instances. In this cases, method like mixed-models GWAS implemented in BOLT may be unsuccessful to fully correct for stratification. I'd recommend that the authors, at least use projected PCs to call ancestry. I think this has been done centrally by the UKB so the authors could request that information.

Replication. The authors propose to assess the replication or replicability of their results by looking up the association within sexes. I don't think this is a proper replication approach and therefore I'd recommend that the authors remove it. They could for example either perform a look up in summary statistics from previous GWAS or they could look up in non-European ancestry participants the UKB. In particular within South-Asian ancestry participant who show the smallest F_{st} (differentiation) with European ancestry participants.

Collider bias. Many of the genes/loci highlighted by the authors seems to overlap with those of height. I'm wondering to which extent this could be due to collider biases given that the authors have adjusted their analyses for height (and height squared), which is an heritable trait. Can the author perform an analysis unadjusted for height and show correlation of effect sizes? Also, check how many of these 717 SNPs remain genome-wide significant when not adjusting for height?

Variance explained. The authors calculate the variance explained by the genome-wide significant SNPs by simply summing up the $2p(1-p)b^2$, where p is the minor allele frequency (MAF) and b is the estimated SNP effect. Because of estimation error (although small here given and sample size is quite large) and most importantly because of winner's curse this estimator of the variance explained is necessarily inflated. The ideal case would be to re-estimate the variance explained in an independent sample. Or at the very least perform a correction for Winner's curse.

MR. The authors applied a test for heterogeneity (HEIDI) to exclude the possibility that their finding are driven by pleiotropy and not by causality. The threshold chosen ($p < 1e-5$; line 311) is too lenient for HEIDI. People use 0.05 if you want to be sure that your results are not driven by pleiotropy. If would suggest this threshold instead.

Minor comments.

Line 58: Despite the large and NOT Despite of

Line 59: percentage of

Line 71: specify here if the >700 SNPs are independent

Line 73: please remove “important” here

Line 93: Put a reference here or point to the method section with more details

Line 117: Put a reference there to justify the threshold (e.g. <https://www.ncbi.nlm.nih.gov/pmc/articles/PMC6351539/> or

<https://genomebiology.biomedcentral.com/articles/10.1186/s13059-017-1216-0>)

Line 122: why guess a reasonable number when you can directly assess how many loci you give based on a certain definition (e.g. that of Wood et al 2014)

Line 124-126: This is not a replication per se.

Line 141: UKB team should have provided groups with a list of related individuals. Have you used that list or did you rerun KING to identify them?

Line 156: Not quite sure where this number comes from?

Line 172: Could you please specify here that you used KING with genotyped data and no imputed?

Line 190: Are these 369,968 identified using KING? Please clarify here.

Line 231: SMR is not a SNP prioritisation method but a GENE prioritisation method.

Line 270: I think that “computational” would sound better here than “running”.

Line 273-274: That looks like a trick to make LDpred run faster, right? Please clarify and justify why you’ve pre-selected these SNPs. In principle LDpred could run without.

Line 303: Please rephrase as the sentence does not make sense.

Line 351: Did you use clumping to identify these loci? Please clarify.

Line 371: Are these 302 SNPs the most associated one vs the 259? The conclusion is a bit rushed otherwise.

Line 381-388: Your set of SNPs is ascertained to have limited differences between sexes. Therefore you lack power to identify sex-specific effects. That should be acknowledged somewhere (discussion probably).

Line 387: please report these effects here.

Line 393: “As expected...” please justify why this relation expected? Do we have evidence of negative selection for this trait?

Line 400: ER (enrichment ratio) would be better than EA.

Line 421: rs400596 is an eQTL for PAM in blood. Do you also have the strength of association with expression of PAM in skeletal muscle? Please report it as that would add to your demonstration.

Line 424: its potential function relevance

Line 433: Please replace “are not able to derive” with “could not be calculated”. Do you know why? Is it because there were not many I suppose.

Line 438-443: $r^2 > 0.3$ is too lenient for rare variants. The main challenge with imputation is allele frequency. So Are the 33 variants common or rare? If the former then, this is expected.

Line 556: Your analyses were adjusted for AFM, then you find a negative correlation with fat mass. This looks like collider to me.

Line 593: single variant do not have heritability. You could instead speak about variance explained by the variant.

Line 797: Could you please specify what curve has been fitted?

Responses to Comments for

" Genetic Architecture of Appendicular Lean Mass Revealed in the UKB Study "

Reviewer #1

Comment #1 *"Study participants. Why the authors select self-reported white individuals? The authors carried out the genetic association analysis using linear mixed models to control for population structure. Furthermore, it is not clear for me why the authors discarded individuals genotyped but no imputed. In total, how many individuals are removed from the initial dataset?"*

Response: Thank you very much for your constructive comments.

Although mixed-model analysis is effective in correcting for population structure, careful data quality control remains critical. Therefore, we only analyzed self-reported white individuals, as recommended by the BOLT-LMM paper [1], as a key first measure to control potential population stratification. Many previous studies also focused on self-reported white individuals, such as [2, 3].

In the UKB analysis, a total of 968 participants were identified with unusually high heterozygosity and/or missing rates [4]. These genotyped participants were not imputed and therefore were excluded from our association analysis.

In total, 487,378 participants were both phenotyped and imputed, of whom, 37,135 were removed. The final number of participants is 450,243. Please refer to Page 6 Lines 97-100.

Comment #2 *"Candidate genes prioritization. Is the PAM gene the only one that has the six sources of evidence? What about the combinations of five, four, three, etc. source of evidences? It could be good if the authors could deep into the results by exploring the sources using for example the `UpsetR` package (<https://cran.rstudio.com/web/packages/UpSetR/vignettes/attribute.plots.html>)"*

Response: In this revised manuscript, we have plotted an intersection figure using UpsetR package per your kind suggestion. Please check **supplemental figure 2**.

Comment #3 *"Mendelian randomization analysis. How many variants were used for the MR analysis for each disease/metabolic trait? It could be nice if the authors could make a Figure to visualize the results, as for example, a scatterplot of the betas and the regression line analyzed for both ALM and each disease/metabolic trait."*

Response: Please refer to **supplemental table 16** for the number of variants used for MR analysis (the "nsnp" column) and **supplemental figure 3** for the scatterplots of the betas.

Comment #4 *"In order to enrich the whole analysis performed in this valuable work the authors could make a flowchart (detailing number of individuals, significant variants, variants used in post GWAS-analysis, etc..) including GWAS for each gender, meta-analysis, conditional analysis, genetic architecture, candidate genes, mendelian randomization analysis, etc."*

Response: We have drawn a flowchart per your kind suggestion. Please refer to **Figure 1**.

Comment #5 *"Line 102. Genotype quality control. How do you phase haplotypes of the UK Biobank individuals? How do you impute variants? You explain that you use three reference panels, what do you do in case that a variant is present in more than one panel? Are the ~19.4 million variants unique?"*

Response: All phasing and genotype imputation were centrally performed by the UKB. The UKB released the imputed genotypes that were used for downstream association analyses in our manuscript. The detailed process of genotype imputation and haplotype phasing was described in the UKB pipeline paper [4].

The 19.4 million variants are unique. Please refer to Page 7 Lines 118-119.

Comment #6 *"Lines 141 and 172. KING software. What threshold of the kinship coefficient do you use to select unrelated individuals in these analyses?"*

Response: We used the "--unrelated --degree 2" option, corresponding to a threshold value of 0.0884. Please refer to Page 8 Line 147.

Comment #7 *"Line 156. "561 identified loci", these loci are identified from the genetic association analysis (Line 109) and using $P < 5e-9$?"*

Response: All the 561 loci had $P < 5e-9$ in the meta-analysis as well as $P < 5e-5$ in each gender. In this revised manuscript, we have re-done the genetic association analyses without adjusting height, in response to Reviewer #2. A total of 1,059 variants were identified.

Comment #8 *"Line 303. I do not find the links to download the GWAS summary statistics."*

Response: In this revised manuscript, we provided the link to download the summary statistics <https://www.ebi.ac.uk/gwas/downloads/summary-statistics>. Please refer to Page 20 Lines 385-387.

Comment #9 *"Line 336. "controlling for AFM". Do you use AFM as covariate in the GWAS*

analysis? If yes, I would add this in Lines 109-114. I understood that AFM was adjusted only in the transformation of ALM as explained in the methods section (Line 88)."

Response: We did not use AFM as a covariate in the GWAS analysis. AFM was only used to adjust ALM in phenotype modeling. AFM was not correlated with the phenotype being analyzed.

Comment #10 *"Lines 371-375. Are these results obtained by comparing with the GIANT Consortium? If yes, please, clarify this."*

Response: These traits were retrieved from the GIANT consortium, for which we have clarified on Page 20 Line 400.

Comment #11 *"Lines 423-424. "PAM is also prioritized by both SMR and DEPICT.." This statement is redundant since at the beginning of the paragraph is mentioned that PAM has the six sources of evidence."*

Response: Removed.

Comment #12 *"Line 445. "Majority of these 42 variants". Please, quantify with the number or percentage of variants."*

Response: We corrected with "Majority (28) of these 51 mis-sense mutations" on Page 24 Line 484.

Comment #13 *"10- Typo in the title of the submitted manuscript. "Appendilar"."*

Response: corrected.

Thank you very much again for your comments!

Reviewer #2

Comment #1 *"Collider bias. Many of the genes/loci highlighted by the authors seems to overlap with those of height. I'm wondering to which extent this could be due to collider biases given that the authors have adjusted their analyses for height (and height squared), which is an heritable trait. Can the author perform an analysis unadjusted for height and show correlation of effect sizes? Also, check how many of these 717 SNPs remain genome-wide significant when not adjusting for height?"*

Response: Thank you very much for your constructive comments.

Per your request, we have performed additional association analyses without adjusting ALM by height and height squared. The results showed that the collider bias was indeed present for a considerable portion of variants. Specifically, the correlation coefficient of effect sizes (regression coefficient) was 0.80. Up to 350 of the 717 lead variants have become insignificant at the GWS level, and 63 of them even became insignificant nominally.

Therefore, to avoid collider bias, we decided not to use height and height square as covariates in this revised manuscript. We noticed that none of the several recent GWAS of lean mass used height as covariate [5-7]. We have re-done all follow-up analyses. In brief, a total of 1,059 variants from 799 loci were identified, explaining 15.5% phenotypic variance.

Comment #2 *"Ancestry. As far I can judge, they authors have determined ancestry based on self-report, which may be unreliable in many instances. In this cases, method like mixed-models GWAS implemented in BOLT may be unsuccessful to fully correct for stratification. I'd recommend that the authors, at least use projected PCs to call ancestry. I think this has been done centrally by the UKB so the authors could request that information."*

Response: The PCs were centrally calculated and released by the UKB. The number of PC determined "Caucasian" participants was 409,616, a sample size that was ~40,000 less than that of self-reported white participants. Though the sample of Caucasian participants was more homogeneous, the reduced sample size may cause a certain loss of statistical power for association testing. In contrast, even with a modest level of population stratification in the self-reported white population, the linear mixed model was still able to correct for it. Indeed, the analysis of self-reported white participants was recommended by the authors of BOLT-LMM, and was performed on 23 UKB traits in the BOLT-LMM publication [1]. We also noticed that multiple previous studies analyzed self-reported white participants, such as [2, 3].

In this revised manuscript, we have performed additional association analysis in 400,879 PC determined Caucasian participants. The attenuation ratio (AR), which measures the relative

contribution of confounding factors and is defined as $(\text{intercept}-1)/(\text{mean } \chi^2-1)$, is 0.13. This measure is equal to that from the self-reported white populations, implying that including self-reported white participants may not introduce additional population stratification. In the 400,879 PC determined Caucasian participants, of the 1,059 lead primary SNPs (as detected in the self-reported white population), up to 980 (92.5%) remain significant at the 5×10^{-9} level. P-values for the remaining 79 SNPs range from 5.09×10^{-9} to 4.56×10^{-6} . The weaker signals are likely due to the reduced sample size instead of population stratification.

Please refer to Page 19 Lines 364-366.

Comment #3 *"Replication. The authors propose to assess the replication or replicability of their results by looking up the association within sexes. I don't think this is a proper replication approach and therefore I'd recommend that the authors remove it. They could for example either perform a look up in summary statistics from previous GWAS or they could look up in non-European ancestry participants the UKB. In particular within South-Asian ancestry participant who show the smallest F_{st} (differentiation) with European ancestry participants."*

Response: We looked up the summary statistics of 3 obesity related traits, BMI, WC and WHR in the GIANT consortium. Of the 1,059 lead variants identified in the present study, 326 are found in the obesity traits summary statistics. About 50 are significant at $p < 0.05$ in each of the 3 traits. This number is larger than expected (~20), but still accounts for only a small portion of total number. We did not performed associations in participants of south Asian (Indian, Pakistani and Bangladeshi) population because of the relatively small sample size (maximal $N=8,254$) compared with our study.

With due respect, we would like to argue that our GWAS replication was indeed rigorous and conservative. Essentially, our study may be performed based on a two-stage design, with the first stage involving GWAS in one sex group (e.g., male), and the second stage replicating top hits in the other sex group (e.g., female). As we assumed a maximal number of 1,000 independent loci for the lean mass trait, we selected the top 1,000 hits from the first stage (male group) for replication in the second stage (in female group). Since we only tested 1,000 independent SNPs in the second stage, a Bonferroni-corrected significance level of 5×10^{-5} ($0.05/1000$) was conservative to declare a successful replication. Indeed, in our original analysis with height adjusted, the number of independent loci with $p < 5 \times 10^{-5}$ was smaller than 1,000 in both sex groups, which means that had we followed a two-stage design all the loci would have passed the significance threshold of 5×10^{-5} and been replicated.

In this revised analysis without height adjustment, the number of independent loci with

$p < 5 \times 10^{-5}$ was 1,988 and 1,713 in female and male populations, respectively, almost twice the presumed number of 1,000. This may inflate type I error rate for the Tier 3 variants (i.e., whose p values were in the range of $5 \times 10^{-5} > p > 5 \times 10^{-9}$). Per your request, we removed the replication statement from the methods and results section. But we discussed our interpretation in the DISCUSSION section. Please refer to Page 31 Lines 629-640.

Comment #4 *"Variance explained. The authors calculate the variance explained by the genome-wide significant SNPs by simply summing up the $2p(1-p)b^2$, where p is the minor allele frequency (MAF) and b is the estimated SNP effect. Because of estimation error (although small here given and sample size is quite large) and most importantly because of winner's curse this estimator of the variance explained is necessarily inflated. The ideal case would be to re-estimate the variance explained in an independent sample. Or at the very least perform a correction for Winner's curse."*

Response: We agree that the SNP effect may be over-estimated due to the winner's curse effect. However, as stated by Zhong and Prentice [8], the winner's curse bias becomes negligible as the sample size becomes large. As you have pointed out, given the huge sample size (total $N=450,000$), the magnitude of the inflation was expected to be very minor in the present study.

In this revised manuscript, we have corrected for the winner's curse effect by a simple FDR based method [9]. The explained phenotypic variance is 17.8% and 15.5% before and after the FDR correction, respectively. We have also estimated the heritability explained by all the identified loci. Specifically, we removed all variants in the identified loci (lead SNP + 500 kb flanking region to either side) from the original data, and estimated the total heritability using BOLT-REML. We then subtracted the estimated heritability from another estimation based on all variants, which produced the heritability explained by the identified loci. As a result, the identified loci collectively explained a heritability of 0.279 and 0.288 in females and males, respectively. Therefore, there may still exist a portion of ~10% heritability in the identified loci, whose contribution to the phenotypic variance is yet to be defined.

Please refer to Page 10 Lines 192-196, Page 11 Lines 200, Pages 21-22 Lines 428-436.

Comment #5 *"MR. The authors applied a test for heterogeneity (HEIDI) to exclude the possibility that their finding are driven by pleiotropy and not by causality. The threshold chosen ($p < 1e-5$; line 311) is too lenient for HEIDI. People use 0.05 if you want to be sure that your results are not driven by pleiotropy. If would suggest this threshold instead."*

Response: Per your request, we have used a threshold of 0.05 for HEIDI test. Please refer to Page

17 Line 334.

Minor comments

Line 58: Despite the large and NOT Despite of

Response: Corrected.

Line 59: percentage of

Response: Corrected.

Line 71: specify here if the >700 SNPs are independent

Response: Specified.

Line 73: please remove "important" here

Response: Removed.

Line 93: Put a reference here or point to the method section with more details

Response: Done. Please refer to reference [10].

Line 117: Put a reference there to justify the threshold (e.g. <https://www.ncbi.nlm.nih.gov/pmc/articles/PMC6351539/> or <https://genomebiology.biomedcentral.com/articles/10.1186/s13059-017-1216-0>)

Response: Done.

Line 122: why guess a reasonable number when you can directly assess how many loci you give based on a certain definition (e.g. that of Wood et al 2014)

Response: This number was used to set significance threshold for individual sex group. We set the number in a prospective manner.

Line 124-126: This is not a replication per se.

Response: We have removed the statement.

Line 141: UKB team should have provided groups with a list of related individuals. Have you used that list or did you rerun KING to identify them?

Response: We re-run the KING to identify a list of unrelated white individuals.

Line 156: Not quite sure where this number comes from?

Response: Removed.

Line 172: Could you please specify here that you used KING with genotyped data and no imputed?

Response: Specified.

Line 190: Are these 369,968 identified using KING? Please clarify here.

Response: In this revised manuscript, PCs were included as covariates in the main analysis, so we did not compare the results with those from unrelated participants.

Line 231: SMR is not a SNP prioritisation method but a GENE prioritisation method.

Response: Corrected.

Line 270: I think that “computational” would sound better here than “running”.

Response: Corrected.

Line 273-274: That looks like a trick to make LDpred run faster, right? Please clarify and justify why you’ve pre-selected these SNPs. In principle LDpred could run without.

Response: Justified. It was mainly for efficient use of computer memory, because LDpred requires a validation sample with original genotypes, which could be huge at the genome-wide scale. Please check Page 15 Line 292.

Line 303: Please rephrase as the sentence does not make sense.

Response: Rephrased.

Line 351: Did you use clumping to identify these loci? Please clarify.

Response: We did not use clumping. Instead, we divide all the SNPs into loci based on their physical coordinates.

Line 371: Are these 302 SNPs the most associated one vs the 259? The conclusion is a bit rushed otherwise.

Response: We checked the flanking regions (500 kb to either direction) surrounding each of these

302+259 lead SNPs for association with obesity traits. These 302 SNPs correspond to 302 regions, where one or more SNPs in each region show association with obesity traits. Whereas no SNPs in the remaining 259 regions shows the association.

Line 381-388: Your set of SNPs is ascertained to have limited differences between sexes. Therefore you lack power to identify sex-specific effects. That should be acknowledged somewhere (discussion probably).

Response: In our study, we tried to differentiate sex specificity from sex heterogeneity. Sex specificity refers to the genetic effect present in one but not in the other sex. Sex heterogeneity refers to the genetic effect present in both sexes, with the effect size different in two sexes.

For sex specificity, our analysis identified 107 loci that are significant ($p < 5 \times 10^{-9}$) in between-sex meta-analysis but not significant at the suggestive level ($p < 5 \times 10^{-5}$) in each sex group. These loci may represent potential sex specificity loci pending further replication.

For sex heterogeneity, of the 1,059 lead variants that are identified in this revised analysis, 181 (17.1%) have a high level between-gender meta-analysis heterogeneity ($I^2 > 50\%$), most (175) of which belong to Tier 1 or 2 variants. A statistical test on sex difference in allele effect size shows no difference for any SNP after accounting for multiple testing ($\alpha = 0.05/1,059 = 4.72 \times 10^{-5}$), suggesting that all of the variants have similar effect sizes between sexes.

Please refer to Pages 20-21 Lines 405-415.

Line 387: please report these effects here.

Response: These effects become non-significant in this revised manuscript.

Line 393: "As expected..." please justify why this relation expected? Do we have evidence of negative selection for this trait?

Response: Our expectation is from a statistical perspective, where a rare allele should have a larger per allele effect in order for it to be identified. In this revised manuscript, we removed "as expected" to avoid misunderstanding.

Line 400: ER (enrichment ratio) would be better than EA.

Response: Corrected.

Line 421: rs400596 is an eQTL for PAM in blood. Do you also have the strength of association

with expression of PAM in skeletal muscle? Please report it as that would add to your demonstration.

Response: The SNP rs400596 is also significant in the association with PAM expression in skeletal muscle tissue ($p=4.24\times 10^{-7}$).

Line 424: its potential function relevance

Response: Rephrased.

Line 433: Please replace “are not able to derive” with “could not be calculated”. Do you know why? Is it because there were not many I suppose.

Response: Replaced. The reason was probably that the independent variable (allele dosage) had limited variation (due to rare variants) so that the linear regression was not applicable.

Line 438-443: $r^2 > 0.3$ is too lenient for rare variants. The main challenge with imputation is allele frequency. So Are the 33 variants common or rare? If the former then, this is expected.

Response: These variants contains both common and rare variants. In this revised manuscript, we have calculated correlation coefficient between imputed and sequenced genotypes. r^2 value ranges from 0.84 to 1.00, which is quite high. Please refer to Page 23 Line 469.

Line 556: Your analyses were adjusted for AFM, then you find a negative correlation with fat mass. This looks like collider to me.

Response: We realized this issue in this revised manuscript. Please refer to Page 28 Lines 579-582.

Line 593: single variant do not have heritability. You could instead speak about variance explained by the variant.

Response: Corrected.

Line 797: Could you please specify what curve has been fitted?

Response: The curve was a power trend line fitted in Excel.

Thank you very much again for your comments.

Reviewer #3

Comment #1: *"Most importantly, the work and the findings need to be put in context of the available literature. An important piece (ALM GWAS in a younger subset of the UK Biobank excluding age related muscle loss) is missing: <https://www.ncbi.nlm.nih.gov/pubmed/31761296> Thus, it is important to carefully relate the findings to those published earlier."*

Response: Thank you very much for your constructive comments. The mentioned paper had not been published at the time of our original manuscript being developed and submitted (May 20, 2019). In this revised manuscript, we have included that important paper. Please refer to Page 5 Lines 70-73.

Comment #2: *"In relation to the publication mentioned above, it would be interesting to see age-specific effects (how do the effect of the discovered SNPs compare in middle-aged vs elderly?)."*

Response: In this revised manuscript, we have performed additional analysis to evaluate age-dependent effect of the identified lead SNPs. Specifically, the sample was divided into the following 6 age strata defined by bins of 5 years: 45 or less (N=54,608), 46-50 (N=58,865), 51-55 (N=70,253), 56-60 (N=89,479), 61-65 (N=10,9696) and 66 or more (N=67,342). Participants of both genders within each age stratum were pooled together for analysis. The association within each age stratum was examined by BOLT-LMM. The generated regression coefficients from all age strata were meta-regressed against mean age of each stratum to examine the effect of age on genetic effect. Meta-regression was implemented by a weighted linear regression analysis weighted by inverse variance of each regression coefficient.

There were 1,059 lead SNPs. At the Bonferroni corrected significance level ($p < 0.05/1,059 = 4.72 \times 10^{-5}$), no age-dependent effect was identified, though dozens of variants are nominally significant.

Please refer to Page 11 Lines 202-213, Page 21 Lines 416-422.

Comment #3: *"Are the authors aware of the issues with the currently released versions of the exome sequencing data from the UK Biobank [systematic under-marking of duplicate reads in the SBP pipeline and under-calling of variants in primary assembly regions from which the alternative contigs are derived for the FE exome dataset.]? How was this handled?"*

Response: Yes, we have noticed the issue, which was announced after we completed the analysis. Since the UKB has not released the corrected dataset, we could only acknowledge this issue in this revised manuscript. Meanwhile, given the extremely high concordance between imputed and

sequenced variants, we believe that this issue may be minor. Please refer to Page 9 Line 165.

Comment #4: *"Genes involved in multiple gene sets are likely to act as hub genes and may play a central regulatory role." While this would be true in an ideal world, the way GO and other gene sets are defined are highly bias towards more studied genes, generating a vicious circle. Also the fact that the 2nd and 3rd most often appearing gene show no significant gene-level score, supporting no link with the disease, may simply represent study-bias."*

Response: Per your kind suggestion, we have removed this paragraph to avoid misleading interpretation.

Comment #5: *"It is also negatively correlated with body fat ($rg=-0.17$, $9.44E-6$), suggesting an opposite developmental direction towards lean and fat mass." – I'm not sure I agree with the interpretation. Any SNP that is responsible for general body size (height or weight) will show an opposite effect on lean and fat mass, because it is associated with something that is related to the sum of these two measures."*

Response: This negative correlation could be either due to phenotypic constraint between the two measures, or due to collider bias as pointed out by reviewer #2, since it was fat mass adjusted lean mass that was used in our association analysis. In this revised manuscript, we re-interpreted the results with caution. Please refer to Page 28 Lines 579-582.

Comment #6: *"As a proof of concept and support of the bioimpedance based measure, could the derived PRS be correlated with the DXA-measured lean mass (in up to 5K samples)? A score that explains 10% of trait variance must have an easily detectable correlation in 4k samples."*

Response: In this revised manuscript, we have performed additional analysis to evaluate the capacity of our findings to predict DXA measured lean mass. Specifically, we re-performed the association analysis in participants who did not undertake the DXA scan, and then derived the PRS for participants who undertook the DXA scan. The correlation coefficient between PRS and DXA derived lean mass was 0.18 (95% CI [0.15, 0.21], $p<2.2\times 10^{-16}$), demonstrating the capacity of the present findings in predicting DXA-measured lean mass. Please refer to Page 15 Lines 297-301, Page 28 Lines 565-568.

Comment #7: *"I wonder whether the authors see any relationship between (self-reported or preferably actimeter-measured) physical activity and lean mass? If so, should it be corrected for?"*

Response: We did not examine the relationship between them. While we acknowledge that

physical activity does have a significant effect on lean mass [11], our requested data did not include any physical activity related items. In this revised manuscript, we acknowledged this as a significant limitation in the discussion section. Please check Page 32 Lines 660-662.

Comment #8: *"While the authors corrected for height and height², I was wondering how similar their trait residuals are to a lean body mass index (=lean mass/(height²)) measure?"*

Response: In our opinion, the adjustment of height and height² is more accurate than lean body mass index in correcting for the effect of height, because lean mass may not have a perfect relationship with height². In response to reviewer #2, we have noticed that including height into covariates has caused collider bias. Therefore, we instead analyzed appendicular lean mass without adjustment of height. We also noticed that several previous GWAS studies did not adjust for height either [5-7].

Comment #9: *"Phenotypic outliers were monitored by the Tukey method." – You should be more specific, have those values been removed? Did you use k=1.5?"*

Response: Yes k=1.5 was used to monitor the outlier. However, these outliers were not removed, because neither fat mass nor lean mass followed a normal distribution. Instead, both distributions were right skewed (Supplementary Figure 1), which means an excess number of outliers were present at the right tail of the distribution. When using the Tukey (k=1.5) method, up to 8,787 male subjects and 10,948 female subjects were classified as outliers. But they were not necessarily extreme observations given such a huge sample size. For example, the range to define normal male lean mass was [18.25, 38.65] kilograms. Any values outside this range would be considered as outliers, which was obviously over conservative. Excluding these outliers may reduce statistical power in subsequent association test. Therefore, we did not exclude these extreme outliers in order to maximize the power for association test.

The most adverse effect caused by extreme outliers is that they may bias the estimation of effect size. However, in our analysis the raw lean mass was already transformed into a standard normal distribution variable that was later used in association test. As a matter of fact, no outliers were observed in the transformed variable that was used for effect size estimation and hence the potential adverse effect of outliers is unlikely to exist in our analysis.

Comment #10: *"Excluding imputation $r^2 < 0.3$ is not stringent enough, usually people keep SNVs with $\hat{r}^2 > 0.7$ or 0.8 How many of your top hits fall in the $0.3 < r^2 < 0.7$ category?"*

Response: Only 18 of 1,059 (1.7%) lead variants fall in the $0.3 < r^2 < 0.7$ category. The minimum

and maximum of them are 0.40 and 0.69, and the average value of them is 0.57. The average r^2 score across all the 1,059 lead variants is as high as 0.97, demonstrating high imputation certainty.

Comment #11: "*“across-gender meta-analysis” -> “between-gender meta-analysis”*"

Response: Changed.

Comment #12: "*“almost all of the identified variants may have similar effect sizes across genders” - It has to be pointed out that this is mainly due to the selection of the 717 SNPs. Those are significant in sex-combined analysis, thus expected to have similar effects between sexes. The way to detect sex-specific associations is to go deeper in sex-combined association P-values, where one has a chance to detect effects specific to one sex only, for example. I understand that sex-specificity is out of the scope of the paper, but the statement needs to be adjusted not to mislead the readers.”*"

Response: In this revised manuscript, we have divided this section into two separate ones: sex specificity effect and sex heterogeneity effect. Sex specificity refers to the genetic effect present in one but not in the other sex. Sex heterogeneity refers to the genetic effect present in both sexes, with the effect size different in two sexes.

For sex specificity, our analysis identified 107 loci that are significant ($p < 5 \times 10^{-9}$) in between-sex meta-analysis but not significant at the suggestive level ($p < 5 \times 10^{-5}$) in each sex group. These loci may represent potential sex specificity loci pending further replication.

For sex heterogeneity, of the 1,059 lead variants that are identified in this revised analysis, 181 (17.1%) have a high level between-gender meta-analysis heterogeneity ($I^2 > 50\%$), most (175) of which belong to Tier 1 or 2 variants. A statistical test on sex difference in allele effect size shows no difference for any SNP after accounting for multiple testing ($\alpha = 0.05/1,059 = 4.72 \times 10^{-5}$), suggesting that all of the variants have similar effect sizes between sexes.

Please check Pages 20-21 Lines 405-415.

Comment #13: "*“Though the 7 rare variants are GWS significant in the whole cohort, they appear to have limited imputed dosage variation in the much smaller exome-sequenced sample...”, still the concordance between imputed and sequenced genotypes can be established. What is the r^2 and D-prime for these 7 SNVs?”*"

Response: The r^2 between imputed versus sequenced genotypes ranges from 0.84 to 1.00 for the seven rare variants. Please check Page 23 Lines 469.

Comment #14: *"Please have the manuscript checked by a native English speaker to avoid sentences like "hence their imputation association p-values are not able to derive""*

Response: Done.

Comment #15: *"3,101 genes are significant at the gene-based GWS level" – This gives very little insight into biology. As sample sizes increase, all genes will become significant (e.g. height GWAS paper in preparation). The importance is their relative order."*

Response: We agree. As shown in the manuscript, the major biological insights did not come from these individual genes. The information of genes significant at the gene-based GWAS level is provided only for readers' information and for interested investigators to replicate our results. As examples, we provided the top 5 genes with the highest significance. Please check Page 26 Lines 538-540.

Thank you very much again for your comments.

References

1. Loh, P.R., et al., *Mixed-model association for biobank-scale datasets*. Nat Genet, 2018. **50**(7): p. 906-908.
2. Cole, J.B., J.C. Florez, and J.N. Hirschhorn, *Comprehensive genomic analysis of dietary habits in UK Biobank identifies hundreds of genetic associations*. Nat Commun, 2020. **11**(1): p. 1467.
3. Strawbridge, R.J., et al., *Genome-wide analysis of self-reported risk-taking behaviour and cross-disorder genetic correlations in the UK Biobank cohort*. Transl Psychiatry, 2018. **8**(1): p. 39.
4. Bycroft, C., et al., *The UK Biobank resource with deep phenotyping and genomic data*. Nature, 2018. **562**(7726): p. 203-209.
5. Hernandez Cordero, A.I., et al., *Genome-wide Associations Reveal Human-Mouse Genetic Convergence and Modifiers of Myogenesis, CPNE1 and STC2*. Am J Hum Genet, 2019. **105**(6): p. 1222-1236.
6. Zillikens, M.C., et al., *Large meta-analysis of genome-wide association studies identifies five loci for lean body mass*. Nat Commun, 2017. **8**(1): p. 80.
7. Medina-Gomez, C., et al., *Bivariate genome-wide association meta-analysis of pediatric musculoskeletal traits reveals pleiotropic effects at the SREBF1/TOM1L2 locus*. Nat Commun, 2017. **8**(1): p. 121.
8. Zhong, H. and R.L. Prentice, *Bias-reduced estimators and confidence intervals for odds ratios in genome-wide association studies*. Biostatistics, 2008. **9**(4): p. 621-34.
9. Bigdeli, T.B., et al., *A simple yet accurate correction for winner's curse can predict signals discovered in much larger genome scans*. Bioinformatics, 2016. **32**(17): p. 2598-603.
10. C.Hoaglin, D., *John W. Tukey and Data Analysis* Statistical Science, 2003. **18**(3): p. 311-318.
11. Distefano, G. and B.H. Goodpaster, *Effects of Exercise and Aging on Skeletal Muscle*. Cold Spring Harb Perspect Med, 2018. **8**(3).

Reviewers' comments:

Reviewer #2 (Remarks to the Author):

I thank the authors for addressing most of my comments, in particular those related to collider biases. However, I still think that the study could be improved as I suspect some remaining uncorrected stratification in their results. The attenuation ratio of 0.13 is large. In the BOLT-LMM paper that the authors refer to the ratio statistics is <0.1 for most the traits. It is only >0.11 for traits like hair colour and tanning ability which are highly genetically stratified between continents. In addition, I still think that their replication strategy is somewhat artificial. My suggestion to look at other ancestries could simply involve estimating the correlation of SNP effects across ancestry. We expect that correlation not to be one but at least it gives a some elements of replication. Based on these elements, and although I acknowledge that the authors have improved their manuscript, I cannot recommend the latter for publication.

Reviewer #3 (Remarks to the Author):

R3-Comment #1: "Most importantly, the work and the findings need to be put in context of the available literature. An important piece (ALM GWAS in a younger subset of the UK Biobank excluding age related muscle loss) is missing: <https://www.ncbi.nlm.nih.gov/pubmed/31761296> Thus, it is important to carefully relate the findings to those published earlier."

Response: Thank you very much for your constructive comments. The mentioned paper had not been published at the time of our original manuscript being developed and submitted (May 20, 2019). In this revised manuscript, we have included that important paper. Please refer to Page 5 Lines 70-73.

R3-Comment1R: I was expecting the authors (on page 19) describing how many of their identified variants are truly novel (not part of the 182 loci identified in the previous study).

R3- Comment #9: "Phenotypic outliers were monitored by the Tukey method." – You should be more

specific, have those values been removed? Did you use $k=1.5$?"

Response: Yes $k=1.5$ was used to monitor the outlier. However, these outliers were not removed, because neither fat mass nor lean mass followed a normal distribution

R3 – Comment9R: I agree that such outliers should not be removed (especially since inverse normal quantile transformation has been applied to the trait). But then what is the point of this sentence? What does "monitor" mean, especially with such an irrelevant value of $k=1.5$, which should not be used for such right skewed distributions.

R3 - Comment 13: ""Though the 7 rare variants are GWS significant in the whole cohort, they appear to have limited imputed dosage variation in the much smaller exome-sequenced sample...", still the concordance between imputed and sequenced genotypes can be established. What is the r^2 and D-prime for these 7 SNVs?"

Response: The r^2 between imputed versus sequenced genotypes ranges from 0.84 to 1.00 for the seven rare variants. Please check Page 23 Lines 469.

R3 - Comment 13R: I thought that the 0.84-1.00 range referred to all 51 variants, not only those 7? It seems weird that you have a good P-value agreement for all but 7 SNVs, but for those 7 SNVs the imputation quality is excellent ($r^2\text{-hat}=0.84\text{-}1.00$), it does not make sense. Is there a misunderstanding here?

Responses to Comments for "Genetic Architecture of Appendicular Lean Mass Revealed in the UKB Study"

Reviewer #2

Comment #1 "I thank the authors for addressing most of my comments, in particular those related to collider biases. However, I still think that the study could be improved as I suspect some remaining uncorrected stratification in their results. The attenuation ratio of 0.13 is large. In the BOLT-LMM paper that the authors refer to the ratio statistics is <0.1 for most the traits. It is only >0.11 for traits like hair colour and tanning ability which are highly genetically stratified between continents."

Response: Thank you very much for your constructive comments.

We estimated attenuation ratio (AR) by running LDSC with default setting. The program commend, as listed at the LDSC website (<https://github.com/bulik/ldsc/wiki>), is

```
ldsc.py
--h2 GWAS.sumstats.gz
--ref-ld-chr eur_w_ld_chr/
--out GWAS
--w-ld-chr eur_w_ld_chr/
```

In response to your concern, we downloaded the GWAS summary statistics of the 23 traits released by the BOLT-LMM paper and estimated AR in these traits using the same commend. We got inflated AR values for all traits than the ones reported in the BOLT-LMM paper. For example, for height trait, we got an AR of 0.116, which is higher than the reported one 0.078.

Upon communicating with the corresponding author Dr. Po-Ru Loh of the BOLT-LMM paper, he recommended a more sophisticated commend that was used in their study,

```
ldsc.py
--h2 GWAS.sumstats.gz
--ref-ld-chr baselineLD_v1.1/baselineLD.
--frqfile-chr 1000G_Phase3_frq/1000G.EUR.QC.
--w-ld-chr 1000G_Phase3_weights_hm3_no_MHC/weights.hm3_noMHC.
--overlap-annot
--print-coefficient
--out GWAS
```

With this refined commend, we estimated a ratio of 0.096 for height trait, which is lower than 0.116, but still could not exactly replicate their reported value 0.078. Dr. Loh attributed the

difference to different versions of LDSC.

For our lean mass trait, the refined AR was estimated to be 0.078 (s.e 0.008), which is comparable to that of most of the 23 traits in the BOLT-LMM study. Therefore, we are confident that the inflated estimation was due to different commands of the LDSC software instead of uncorrected population stratification.

In this revised manuscript, we have updated our results. Please check Page 18 Line 367, Page 19 Line 373-375.

Comment #2 *"In addition, I still think that their replication strategy is somewhat artificial. My suggestion to look at other ancestries could simply involve estimating the correlation of SNP effects across ancestry. We expect that correlation not to be one but at least it gives a some elements of replication. Based on these elements, and although I acknowledge that the authors have improved their manuscript, I cannot recommend the latter for publication."*

Response: Upon your request, we have replicated associations in 7,452 participants of South Asian (Indian, Pakistani and Bangladeshi) population of the UKB cohort. At the stringent significance level 4.72×10^{-5} (0.05/1,059), only 1 SNP rs4338565 ($p=2.30 \times 10^{-5}$) was significant, which was not unexpected given the limited replication sample size. At the nominal level 0.05, 124 SNPs were significant. Overall, the effect direction at 767 SNPs was consistent between the two analyses (binomial $p < 2.2 \times 10^{-16}$). The correlation coefficient of regression coefficient was 0.27 (95% CI [0.22, 0.33], $p=1.86 \times 10^{-19}$).

Please check Page 9 Lines 162-169, Page 20 Lines 395-402.

Thank you very much again for your comments.

Reviewer #3

Comment #1: *"I was expecting the authors (on page 19) describing how many of their identified variants are truly novel (not part of the 182 loci identified in the previous study)."*

Response: Thank you very much for your constructive comments. Of the 1,059 identified variants, 189 reside in genomic regions reported by previous smaller studies, while the remaining 870 reside in novel genomic regions. We have added the novelty information. Please check Page 20 Line 413 and Supplementary data 2.

Comment #2: *"I agree that such outliers should not be removed (especially since inverse normal quantile transformation has been applied to the trait). But then what is the point of this sentence? What does "monitor" mean, especially with such an irrelevant value of $k=1.5$, which should not be used for such right skewed distributions."*

Response: Removed upon your suggestion.

Comment #3: *"I thought that the 0.84-1.00 range referred to all 51 variants, not only those 7? It seems weird that you have a good P-value agreement for all but 7 SNVs, but for those 7 SNVs the imputation quality is excellent ($r^2\text{-hat}=0.84-1.00$), it does not make sense. Is there a misunderstanding here?"*

Response: Thank you for pointing out this error. The 0.84-1.00 range referred to the 7 rare variants (supplemental table 2). The PLINK software outputted "NA" for the 7 rare variants. We thought it was a mathematic issue of the regression model due to limited variation at the SNP side. Upon your kind notice, we extracted original data and performed association with R, and got results for all the 7 variants. Again, all p-values were close to sequencing based p-values. The failure of PLINK in outputting results for these variants might be due to a default MAF cutoff (0.01) being set in dosage-based association, though we did not find relevant documentation in PLINK website.

In this revised manuscript, we have updated these results. Please check Page 24 Lines 487-492.

Thank you very much again for your comments.

REVIEWERS' COMMENTS:

Reviewer #2 (Remarks to the Author):

I thank the authors for investigating the stratification issue further. I'm happy with this as long as it is mentioned that some residual stratification may remain uncorrected in their analysis because ancestry was only based upon self-report.

I also thank the authors for showing that some of their findings replicate in other ancestries. I think this is a nice addition to the paper.

I have no more comments.

Responses to Comments for

"The genetic architecture of appendicular lean mass characterized by association analysis in the UK Biobank study"

Reviewer #2

Comment #1 *"I thank the authors for investigating the stratification issue further. I'm happy with this as long as it is mentioned that some residual stratification may remain uncorrected in their analysis because ancestry was only based upon self-report"*

Response: Thank you very much for your constructive comments. Done per your kind suggestion. Please check Page 7 Line 116.

Thank you very much again for your comments.